# A conserved and regulated mechanism drives endosomal Rab transition

**Lars Langemeyer[1], Ann-Christin Borchers[1], Eric Herrmann[2], Nadia Füllbrunn[1], Yaping Han[1], Angela Perz[1], Kathrin Auffarth[1], Daniel Kümmel[2], Christian Ungermann[1,3]***

[1]University of Osnabrück, Department of Biology/Chemistry, Biochemistry section, Osnabrück, Germany; [2]University of Münster, Institute of Biochemistry, Münster, Germany; [3]University of Osnabrück, Center of Cellular Nanoanalytics (CellNanOs), Osnabrück, Germany

**Abstract** Endosomes and lysosomes harbor Rab5 and Rab7 on their surface as key proteins involved in their identity, biogenesis, and fusion. Rab activation requires a guanine nucleotide exchange factor (GEF), which is Mon1-Ccz1 for Rab7. During endosome maturation, Rab5 is replaced by Rab7, though the underlying mechanism remains poorly understood. Here, we identify the molecular determinants for Rab conversion in vivo and in vitro, and reconstitute Rab7 activation with yeast and metazoan proteins. We show (i) that Mon1-Ccz1 is an effector of Rab5, (ii) that membrane-bound Rab5 is the key factor to directly promote Mon1-Ccz1 dependent Rab7 activation and Rab7-dependent membrane fusion, and (iii) that this process is regulated in yeast by the casein kinase Yck3, which phosphorylates Mon1 and blocks Rab5 binding. Our study thus uncovers the minimal feed-forward machinery of the endosomal Rab cascade and a novel regulatory mechanism controlling this pathway.

**\*For correspondence:**
cu@uos.de

**Competing interests:** The authors declare that no competing interests exist.

## Introduction

In any eukaryotic cell the endolysosomal pathway is regulated by two members of the Rab subfamily of small GTPases, namely Rab5 and Rab7 (*Huotari and Helenius, 2011*; *Wandinger-Ness and Zerial, 2014*; *Langemeyer et al., 2018a*). Endocytic vesicles carrying cargo molecules pinch off at the plasma membrane and fuse in a Rab5-dependent manner with early endosomes. Early endosomes undergo fusion with other early endosomes, which in the end mature into late endosomes. All these organelles carry Rab5 as an identity marker. On the verge to late endosomes the identity changes from a Rab5-positive to a Rab7-positive compartment (*Rink et al., 2005*; *Poteryaev et al., 2010*). This allows fusion with the lysosome, where endocytic cargo is degraded.

Rab GTPases cycle between a GDP-bound (Rab-GDP) and a GTP-bound (Rab-GTP) state (*Hutagalung and Novick, 2011*; *Barr, 2013*; *Goody et al., 2017*). To switch between these two forms, they need regulatory proteins. Rab-specific Guanine Nucleotide Exchange Factors (GEF) bind to the Rab GTPase and stabilize the nucleotide-free form to allow dissociation of bound nucleotide and binding of the more abundant GTP. GTPase-Activating Proteins (GAP) are needed to complete the active site of Rab GTPases to hydrolyze bound GTP to GDP. Only when bound to GTP the Rab GTPase can interact with specific effector proteins. Effectors then function as tethering factors, motor proteins or localize specific enzymes, thereby giving a functional identity to organelles.

Rab GTPases are C-terminally prenylated and can thus bind membranes (*Goody et al., 2017*). They are kept soluble in the cytosol by binding in their GDP-form to the Rab Escort Protein (REP) in the context of their posttranslational prenylation or to the Guanine Nucleotide Dissociation Inhibitor (GDI) as their general Rab chaperone. Prenylated Rab GTPases then sample randomly membranes in the cell by dissociating from the chaperone. This process has been postulated to be favored by GDI

Displacement Factors (GDF), hence providing another layer of regulation (*Dirac-Svejstrup et al., 1997*; *Calero and Collins, 2002*; *Sivars et al., 2003*; *Chen et al., 2004*; *Heidtman et al., 2005*). As long as Rab GTPases are not activated, GDI is able to extract the Rab-GDP again. When the Rab GTPase encounters its cognate GEF on a membrane, it is activated by nucleotide exchange and can then bind effectors. The cycle completes, when the corresponding GAP triggers GTP-hydrolysis, and GDI then removes the Rab from the membrane.

Within the endolysosomal system, Rab5 marks early endosomes, whereas Rab7 is found on late endosomes, autophagosomes, and lysosomes (*Chavrier et al., 1990*; *Bucci et al., 1992*; *Wichmann et al., 1992*; *Horazdovsky et al., 1994*; *Gutierrez et al., 2004*; *McEwan et al., 2015*; *Hegedűs et al., 2016*; *Gao et al., 2018*). The Rab5 family has four members in human cells (Rab5a, 5b, 5c, 5d), one member in *Drosophila melanogaster* (RAB5), and at least three members in *S. cerevisiae* (Vps21, Ypt52, Ypt53). In human cells, Rab5 is activated by its GEF Rabex-5 in complex with the Rab5 effector Rabaptin5, which function together in a positive feedback loop to form a Rab5-domain on endosomes (*Wandinger-Ness and Zerial, 2014*; *Franke et al., 2019*). In yeast, at least three Rab5-GEFs have been identified, which may function similarly (*Burd et al., 1996*; *Paulsel et al., 2013*; *Cabrera et al., 2013*; *Bean et al., 2015*). We and others identified the Mon1-Ccz1 complex as the Rab7 GEF (*Nordmann et al., 2010*; *Gerondopoulos et al., 2012*). In yeast, Mon1-Ccz1 forms a dimer, whereas metazoan cells have a third subunit, named RMC1 in mammals (*Vaites et al., 2018*), and Bulli in *Drosophila melanogaster* (Dehnen et al., submitted).

The Rab5-to-Rab7 transition in the endolysosomal pathway is thought to work as a so called Rab-cascade (*Del Conte-Zerial et al., 2008*; *Hutagalung and Novick, 2011*; *Barr, 2013*; *Pfeffer, 2013*; *Langemeyer et al., 2018a*). According to prevailing models, Mon1-Ccz1 is an effector of Rab5, and interactions have been shown by yeast-two- and three-hybrid studies and in pulldown experiments from lysates (*Kinchen and Ravichandran, 2010*; *Cui et al., 2014*). Furthermore, Mon1-Ccz1 also interacts with phosphatidylinositol-3-phosphate, PI-3-P (*Cabrera et al., 2014*; *Lawrence et al., 2014*; *Hegedűs et al., 2016*), which is present on endosomes and autophagosomes (*Schu et al., 1993*; *Kihara et al., 2001*), and functions on endosomes (*Yasuda et al., 2016*). Moreover, it was shown that Mon1/Sand1 alone can displace the Rab5 GEF Rabex-5 from membranes, thus promoting Rab5 release (*Poteryaev et al., 2010*). A similar cascade of a Rab5 to Rab7 transition has been observed on mitochondria in vivo during Parkin-induced mitophagy (*Yamano et al., 2018*). Also here, Mon1-Ccz1 inactivation impaired Rab7 recruitment. Finally, Mon1-Ccz1 binds the LC3-like Atg8 protein and can thus recruit Ypt7 to the yeast autophagosomal membrane (*Gao et al., 2018*).

Despite the evidence that Mon1-Ccz1 can interact with Rab5 and the consecutive order of Rab5 to Rab7 transition on endosomal membranes (*Rink et al., 2005*; *Poteryaev et al., 2010*), there is a lack of mechanistic understanding of this process. Mon1-Ccz1 is obviously key to the Rab5-to-Rab7 transition, but is it also sufficient to drive this process? Is binding to both Rab5-GTP and PI-3-P required for membrane binding and activity?

To address these questions in detail, we reconstituted the Rab5-to-Rab7 transition in vitro by using prenylated Rab5 and Rab7 as soluble factors in complex with their chaperones REP and GDI, and liposomes to mimic the in vivo situation (*Langemeyer et al., 2018b*). We now show that prenylated Rab5 on these membranes is necessary and sufficient to drive Mon1-Ccz1 dependent nucleotide exchange on prenylated Rab7, and subsequently membrane fusion – both in yeast and metazoan cells. In yeast, this process is strongly inhibited and thus regulated by the casein kinase 1-mediated phosphorylation of Mon1. We thus provide an important step in the mechanistic understanding of the endosomal Rab cascade and thus the elucidation of the fundamental principles and regulatory circuits underlying organelle maturation in general.

## Results

### Rab5 is necessary for Mon1-Ccz1 function in vivo

To address the role of PI-3-P and Rab5-like proteins for Mon1-Ccz1 targeting to endosomes or vacuoles, we used the lipophilic dye FM4-64, which is transported in yeast via the endocytic pathway toward the vacuole (*Vida and Emr, 1995*). Wild-type cells take up FM4-64 readily, which results in staining of large vacuoles (*Figure 1A,B*). In contrast, in mutants lacking *mon1* less than 10% of the cells showed a normal vacuolar morphology. Cells deleted for the endosomal Vps30 subunit of the

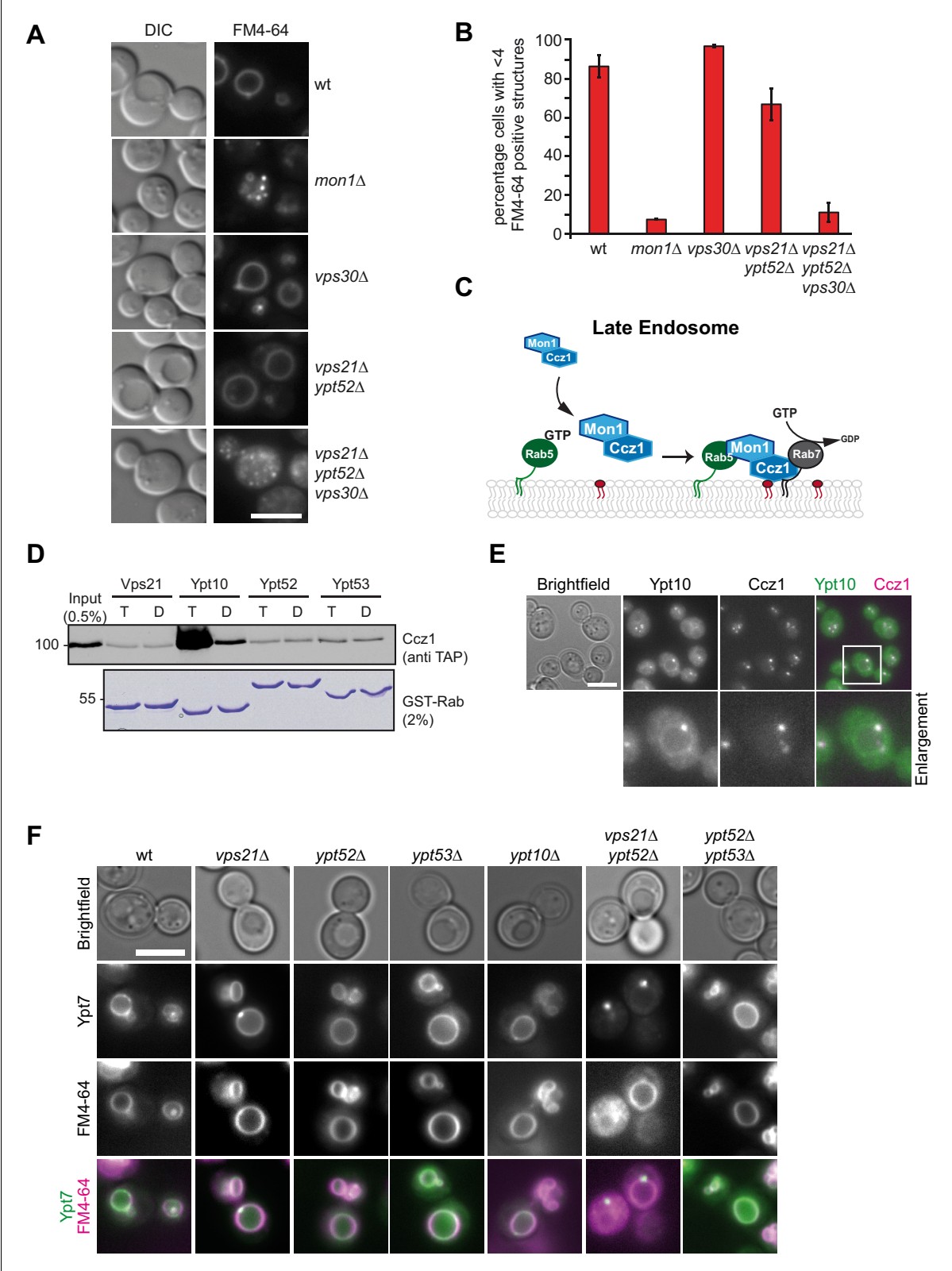

**Figure 1.** Rab5 effect on Mon1-Ccz1 function. (**A**) Vacuole morphology in Rab5 mutants. Cells with the indicated mutations were grown in the presence of 10 μM FM4-64 and analyzed by fluorescence microscopy. Size bar, 5 μm. (**B**) Quantification of vacuole morphology. Percentage of cells with less than four vacuoles is shown. Error bars represent standard deviation. (**C**) Model of cooperation of Rab5-GTP and PI-3-P for Mon1-Ccz1 recruitment to late endosomes. PI-3-P is indicated as red lipid. (**D**) Interaction of yeast Rab5-like proteins with Mon1-Ccz1. Purified GST-tagged Rab5 proteins (Vps21,

*Figure 1 continued on next page*

*Figure 1 continued*

Ypt10, Ypt52, and Ypt53) were loaded with GTP (T) or GDP (D) and incubated with purified Mon1-Ccz1 complex. Eluates were analyzed on SDS-PAGE by Western blotting with an antibody against the TAP-tag on Ccz1 (top) and Coomassie staining (bottom). For details see methods. (E) Localization of Ypt10 in yeast. Cells expressing endogenously GFP-tagged Ypt10 and mKate-tagged Ccz1 were analyzed by fluorescence microscopy. Size bar, 5 μm. (F) Analysis of Ypt7 localization and vacuole morphology in Rab5 deletion strains. mNeon-tagged Ypt7 was expressed under the control of the Ypt7 promoter in cells the indicated Rab5 proteins. Cells were stained with FM4-64, and analyzed by fluorescence microscopy. Size bar, 5 μm.

The online version of this article includes the following source data and figure supplement(s) for figure 1:

**Source data 1.** Quantification of vacuole number in FM4-64 stained wild-type and mutant strains in *Figure 1A*.
**Figure supplement 1.** Vacuole morphology in Rab5 mutants.
**Figure supplement 2.** Interaction of yeast Rab-GTPases with Mon1-Ccz1.

PI-3-Kinase complex II (*Kihara et al., 2001*), or two members of the Rab5-family, Vps21 and Ypt52, which upon deletion impair CORVET or retromer targeting (*Cabrera et al., 2013*; *Paulsel et al., 2013*), had wild-type like vacuoles. Similarly, combinations of different knockouts of members of the Rab5-family did not impair overall vacuole morphology (*Figure 1—figure supplement 1*). However, deleting Vps30, Vps21 and Ypt52 together interfered with a normal endolysosomal maturation and fusion process and resulted in fragmented vacuoles as observed in *mon1Δ* cells (*Figure 1B*). These observations agree with the current model of Rab7 activation on late endosomes, which suggests that Mon1-Ccz1 is brought to the endosomal membrane jointly by active Rab5 or the interaction with PI-3-P (*Figure 1C*; *Langemeyer et al., 2018b*).

Previous analyses used either yeast two-hybrid or pull-down analyses from lysates to conclude that Mon1-Ccz1 is an effector of Rab5-GTP (*Kinchen and Ravichandran, 2010*; *Cui et al., 2014*). To

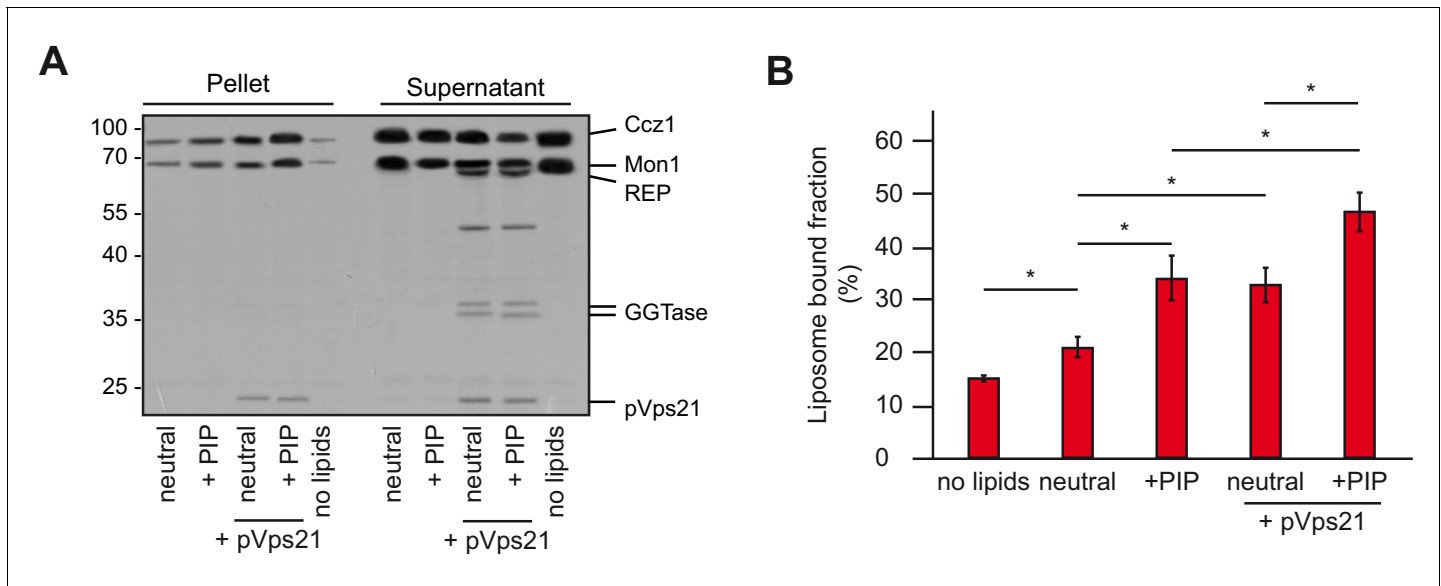

**Figure 2.** Membrane recruitment of Mon1-Ccz1. (A) Liposome binding assay. Liposomes were either composed of 82% PC, 18% PE (neutral), or 79% PC, 18% PE, 2% PI-3-P and 1% PI-3,5-P$_2$ (PIP). Where indicated, prenylated Vps21, which was initially in complex with REP, was activated in the presence of liposomes. Mon1-Ccz1 was added to liposomes, and incubated at room temperature for 20 min. Liposomes were separated by centrifugation into pellet and supernatant (20 min, 20,000 *g*, 4°C), the supernatant was acetone-precipitated. Proteins in pellet and supernatant fraction were analyzed by SDS-PAGE and Coomassie staining. (B) Quantification of pellet fractions from three independent assays. Error bars represent standard deviation. Significance analysis were performed by two-tailed heteroscedastic t-test statistics (*, p≤0.05).

The online version of this article includes the following source data and figure supplement(s) for figure 2:

**Source data 1.** Quantification of *Figure 2A*.
**Figure supplement 1.** Membrane recruitment of Mon1-Ccz1.
**Figure supplement 1—source data 1.** Quantification of *Figure 2—figure supplement 1*.
**Figure supplement 2.** Membrane recruitment of Mon1-Ccz1.
**Figure supplement 2—source data 1.** Quantification of *Figure 2—figure supplement 2*.

test for direct interaction, we purified 9 out of the 11 yeast Rab-GTPases fused to a C-terminal GST-tag. Rabs were loaded with GDP or GTP, immobilized on GSH-Sepharose, and incubated with purified Mon1-Ccz1 complex (*Figure 1—figure supplement 1*). We then probed eluates from beads for bound Mon1-Ccz1 by using an antibody against the TAP-tag on Ccz1. Surprisingly, we only detected weak interactions with most of the tested Rab-GTPases (*Figure 1—figure supplement 1*), including either one of the well-studied Rab5-family members Vps21, Ypt52, or Ypt53. However, we observed a strong interaction with Ypt10-GTP (*Figure 1D*). This poorly characterized Rab-GTPase has been assigned to the Rab5-family previously (*Buvelot Frei et al., 2006*; *Lo et al., 2011*), and localizes to FM4-64 positive structures in yeast Ypt10 overexpression interferes with the endolysosomal system (*Louvet et al., 1999*), suggesting that the protein functions as a Rab5 protein under special growth conditions as shown for Ypt53 (*Paulsel et al., 2013*; *Schmidt et al., 2017*). Indeed, the endogenously expressed GFP-tagged Ypt10 and Ccz1-mCherry colocalized in dot-like structures proximal to the vacuole (*Figure 1E*). We therefore took advantage of Ypt10 in further assays as the strongest Mon1-Ccz1 interactor among yeast Rab5-like proteins.

We next asked if localization of Ypt7 to vacuoles requires indeed Rab5-like proteins, and thus localized functional mNEON-tagged Ypt7 in several deletion strains (*Figure 1F*). Interestingly, none of the single Rab5 deletions showed altered Ypt7 localization. However, deletion of both Vps21 and Ypt52 resulted in a dot-like localization of Ypt7, whereas FM4-64 still stained round vacuoles (*Figure 1A,F*). This suggests that Rab5-proteins can affect Ypt7 localization in vivo.

## Mon1-Ccz1 membrane association requires charged lipids and Rab5 proteins

The Mon1-Ccz1 complex can interact with activated Rab5 (*Kinchen and Ravichandran, 2010*; *Cui et al., 2014*) and associates with membranes containing PI-3-P (*Cabrera et al., 2014*; *Lawrence et al., 2014*). To test both factors for their influence on the membrane association of Mon1-Ccz1, we incubated purified Mon-Ccz1 with liposomes of different composition in the absence or presence of prenylated Rab5-GTP. After allowing for membrane binding, liposomes were recovered by centrifugation and probed for the presence of Mon1-Ccz1 (*Figure 2A*). Using a neutral lipid mix, 21% of Mon1-Ccz1 was associated with the membrane fraction, which increased to 34% in the presence of PI-3-P and PI-3,5-P$_2$. When prenylated Vps21 was included, the membrane association of Mon1-Ccz1 increased further (46%, *Figure 2B*), comparable with Mon1-Ccz1 membrane association in the presence of prenylated Ypt10 (*Figure 2—figure supplement 2*). This suggests that Mon1-Ccz1 takes advantage of a dual targeting mechanism to bind membranes via lipids and Rab5. We also observed comparable binding to membranes carrying other charged lipids at concentrations adjusted for equimolar negative charge (*Figure 2—figure supplement 2*). Thus, Mon1-Ccz1 seems to interact with phospholipids via unspecific electrostatic interactions rather than specific lipid head group recognition.

## Control of Mon1-Ccz1 GEF activity by Rab5 is evolutionarily conserved

To analyze how Mon1-Ccz1 GEF activity is regulated in the context of membranes, we relied on a previously established assay, in which we analyzed the activity of in vitro prenylated Rabs in the context of liposomes (*Thomas and Fromme, 2016*; *Langemeyer et al., 2018b*; *Figure 3A*). We loaded prenylated Ypt7 with the fluorescent GDP analog MANT-GDP, generated the complex with GDI, and measured nucleotide exchange in the presence of purified Mon1-Ccz1 (*Figure 3B,C*). For this, liposomes were incubated with pYpt7:GDI complex in the presence of increasing Mon1-Ccz1 amounts. Surprisingly, we did not observe any GEF activity using liposomes composed of a vacuolar mimicking lipid mix (VML-liposomes) (*Figure 3B*, no recruiter), indicating that recruitment of Mon1-Ccz1 to membranes via charged lipids is not sufficient for GEF activity.

We then preloaded liposomes with various prenylated Rab-GTPases, which were initially complexed by REP (*Figure 3—figure supplement 1*), in the presence of GTP and EDTA (*Figure 3A*). When we now added pYpt7:GDI, Mon1-Ccz1 efficiently triggered nucleotide exchange of Ypt7 in the presence of the Rab5-like protein Vps21, but not the Golgi-Rab Ypt31 (*Figure 3B,D*). Even more activity was observed, when the Rab5-homologs Ypt52 or Ypt53 were present on membranes, and prenylated Ypt10 resulted in strongest GEF activity (*Figure 3B,D*), whereas addition of soluble Ypt10 did not result in Mon-Ccz1 activation (*Figure 3—figure supplement 2*). To ask if the

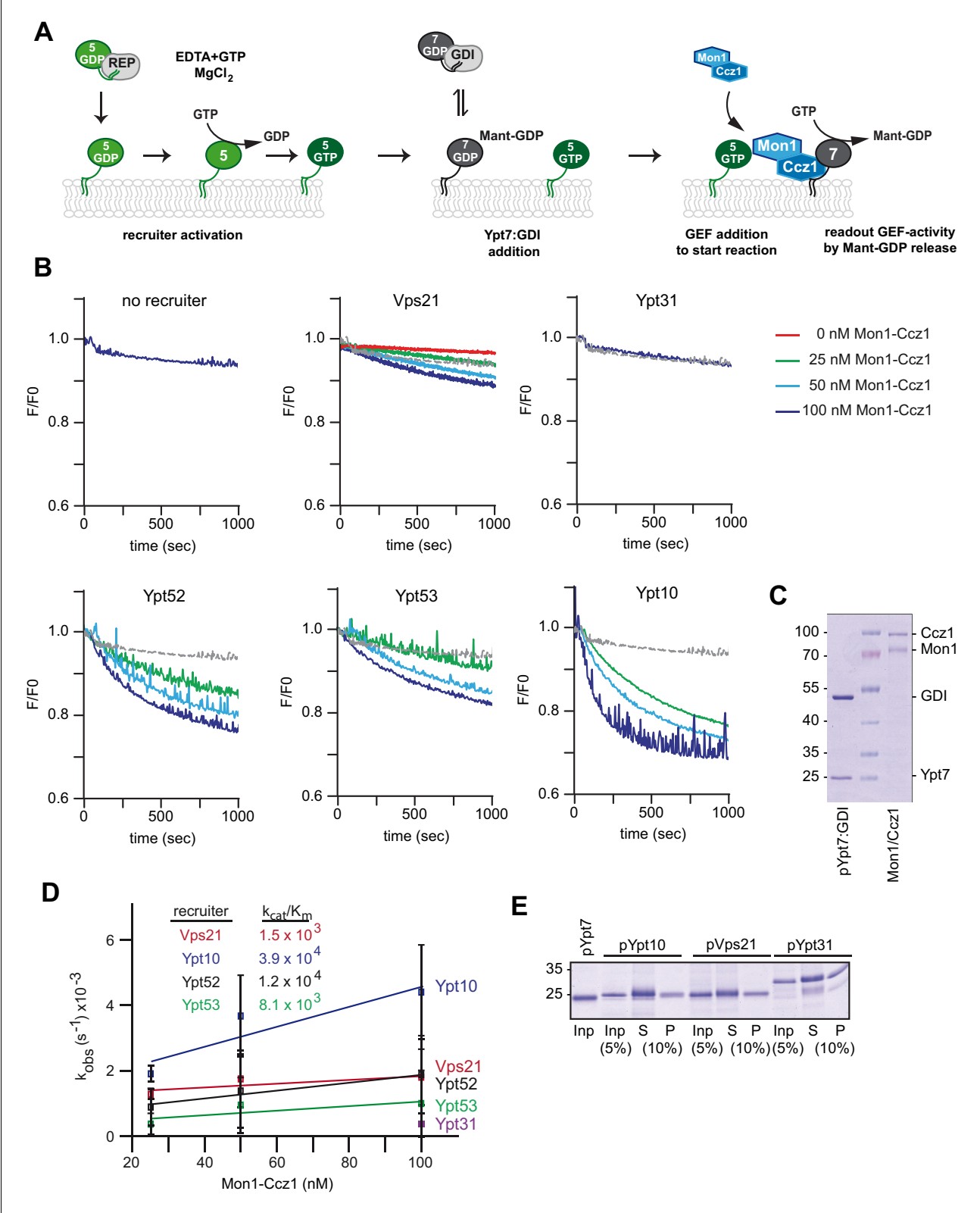

**Figure 3.** Rab5 proteins can activate the Mon1-Ccz1 GEF on membranes. (**A**) Scheme of Rab5-dependent Rab7 activation. Rab5-proteins in complex with the REP were chemically activated to bring protein to membranes. Addition of MANT-GDP-loaded Ypt7:GDI was followed by Mon1-Ccz1 addition. For details see text. (**B**) Recruiter GEF-assay of Mon1-Ccz1. 250 nM MANT-GDP-loaded Ypt7 activation was followed by change in fluorescence over time. Liposomes composed of a vacuolar mimicking lipid mix were incubated with 1.5 µM recruiter GTPase as indicated or buffer as a control (no

*Figure 3 continued on next page*

*Figure 3 continued*

recruiter). For activation of the recruiter, 200 μM GTP was added in the presence of 1.5 mM EDTA for 15 min at room temperature. Afterwards the EDTA was quenched by addition of 3 mM MgCl$_2$. After pre-activation of the recruiter, 250 nM prenylated Ypt7:GDI was added. Increasing amounts of Mon1-Ccz1 (as indicated) were added to start the reaction. Trace of no recruiter sample was plotted into all graphs for reference (grey line) (C) Protein complexes used for recruiter assay. Prenylated Ypt7 in complex with GDI and heterodimeric Mon1-Ccz1 complex were purified as described in the method section, and analyzed by SDS-PAGE and Coomassie staining. (D) Enzymatic parameters as derived from the recruiter assays in (B). Values were calculated from at least two independent measurements. Error bars represent standard deviation. For details see methods. (E) Membrane association of Rab-GTPases in the recruiter-assay. Samples from the recruiter-assay were recovered after 1000 s, and soluble and membrane fraction were separated by centrifugation for 20 min at 20,000 *g*. Fractions were analyzed by SDS-PAGE and Coomassie staining.

The online version of this article includes the following figure supplement(s) for figure 3:

**Figure supplement 1.** Prenylation of yeast Rab-GTPases for the Recruiter GEF-assay.
**Figure supplement 2.** Membrane bound but not soluble Ypt10 activates Mon1-Ccz1.

difference in activity is due to the amount of recruiter GTPase on membranes, we recovered membranes after the measurements, and did not detect any difference between Vps21, Ypt31, and Ypt10 amounts on membranes (*Figure 3E*). These data show that Mon1-Ccz1 dependent activation of the Rab7-like Ypt7 is strongly stimulated by Rab5-GTP on membranes and correlates with the affinity of the effector interaction between Rab and Mon1-Ccz1.

We next asked if the Rab5-dependent Mon1-Ccz1 activation is conserved in metazoans. Mammalian and *Drosophila* Rab7-GEF-complex has a third subunit next to Mon1 and Ccz1, termed RMC1 for regulator of Mon1-Ccz1 (*Vaites et al., 2018*) (Dehnen et al., submitted) (*Figure 4A*). We recently observed no difference in GEF activity for *Drosophila* Rab7-GEF in the absence or presence of the third subunit, named Bulli/CG8270 (Dehnen et al., submitted) in solution. Therefore, we analyzed the dimeric and trimeric (Bulli-) Mon1-Ccz1 complex in the newly established recruiter-GEF-assay. We successfully established the prenylation system for *Dm*Rab5 and *Dm*Rab7 (*Figure 4B*, *Figure 4— figure supplement 1*), purified dimeric and trimeric *Drosophila* (Bulli-) Mon1-Ccz1 complexes (*Figure 4C*, *Figure 4—figure supplement 1*), and measured their activities in the recruiter GEF-assay. Using either complex, we observed efficient GEF-activity on VML-liposomes in the presence of *Dm*pRab5 (*Figure 4D*). GEF-activity of the trimeric GEF was roughly comparable to the dimeric GEF (*Figure 4D,E*), in agreement with their activity in solution using non-prenylated Rabs (Dehnen et al., submitted). We did not observe a difference in the membrane association of the recruiter (*Figure 4—figure supplement 1I*). Notably, also in the *Drosophila* system, we did not detect GEF-activity of either complex in the absence of a recruiter GTPase (*Figure 4F*).

Initially, Bulli/CG8270 was identified as a putative Rab5-effector (*Gillingham et al., 2014*), and thus might support binding of the complex to Rab5. If this were the case, then lower concentrations of recruiter on the membrane may reveal differences in GEF activity. However, lower concentrations of pRab5 in the recruiter assay resulted in the same activity of dimer and trimer as reflected by the similar rate constant (*Figure 4F,G*). These data show that the ability of Rab5-dependent activation requires just the Mon1-Ccz1 core components – in metazoan and yeast. Moreover, we find no evidence that the third subunit in the metazoan complex has a direct influence on the catalytic activity of Mon1-Ccz1. Our data suggest that the interaction of Rab5 with Bulli/CG8270 (*Gillingham et al., 2014*) is indirect and mediated via Mon1-Ccz1.

## Rab5 stimulates Mon1-Ccz1 activity and Rab7-driven membrane fusion

We showed before that fusion of yeast SNARE-decorated proteoliposomes requires just four soluble factors: a soluble SNARE (Vam7), Mon1-Ccz1, the Rab7-GTPase (pYpt7:GDI), and the HOPS tethering complex (*Langemeyer et al., 2018b*; *Figure 5A*). Mon1-Ccz1 activity was limiting for Ypt7 recruitment and activation and thus HOPS function (*Langemeyer et al., 2018b*). As Rab5 can promote Rab7 activation, we expected that a Rab5-like protein could also trigger the membrane fusion machinery in vitro. We reasoned that the presence of a recruiter GTPase such as pYpt10 should lower the demand of Mon1-Ccz1 to initiate the fusion process. First, we titrated Mon1-Ccz1 into the fusion reaction, and detected only minimal fusion at 12.5 nM (*Figure 5—figure supplement 1A,B*). However, in the presence of the recruiter pYpt10, this fusion strongly increased (*Figure 5B*). When we tested the membrane fraction after fusion for the presence of Mon1-Ccz1, we observed comparable amounts of Ccz1 in the membrane fraction. In contrast, the Ypt7-effector Vps41 as a subunit of

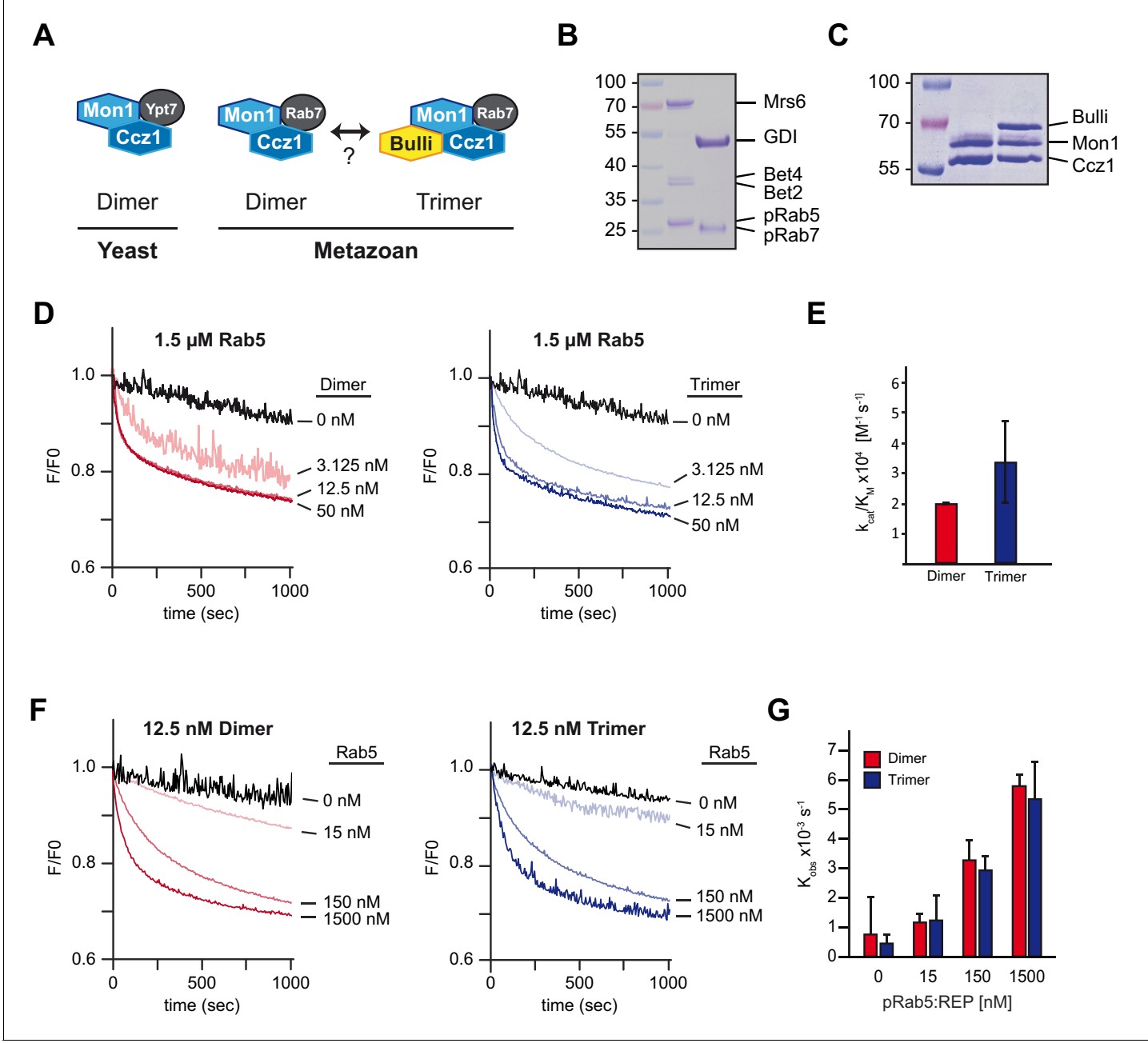

**Figure 4.** Mon1-Ccz1 activation by Rab5 is conserved in metazoan cells. (A) Schematic representing the architecture of yeast and metazoan Rab7-GEF-complexes. Yeast Mon1-Ccz1 consists of a heterodimer whereas metazoan GEF-complex is a heterotrimer with Bulli (in *Drosophila*) or RMC1 in mammalian cells as a third subunit. A dimer of metazoan Mon1-Ccz1 may exist as well. (B) Purification and prenylation of *Drosophila* Rab-GTPases. Rab5 and Rab7 were prenylated and complexed with REP and GDI, respectively. (C) Purification of *Drosophila* dimeric and trimeric Rab7-GEF-complexes from insect cells as used for GEF-assays. (D) Recruiter GEF-assay with *Drosophila* dimeric and trimeric (Bulli-)Mon1-Ccz1. The recruiter GEF-assay was performed as described in *Figure 3* using *Drosophila* protein complexes and increasing amounts of either Mon1-Ccz1 (Dimer) or Bulli-Mon1-Ccz1 (Trimer). (E) Enzymatic parameters as measured in (D). Values were calculated of two independent experiments. Error bars represent standard deviation. (F) Recruiter GEF-assay with decreasing amounts of Rab5. The recruiter-assay was performed as described above with 12.5 nM Mon1-Ccz1 and Bulli-Mon1-Ccz1, respectively, and the indicated amounts of recruiter Rab5. (G) Rate constants as derived from (F) in dependence of concentration of Rab5. Error bars represent standard deviation. For details see Methods.

The online version of this article includes the following source data and figure supplement(s) for figure 4:

**Source data 1.** Quantification *Figure 4E*.
**Source data 2.** Quantification *Figure 4G*.
**Figure supplement 1.** Purification and prenylation of *Drosophila* proteins.

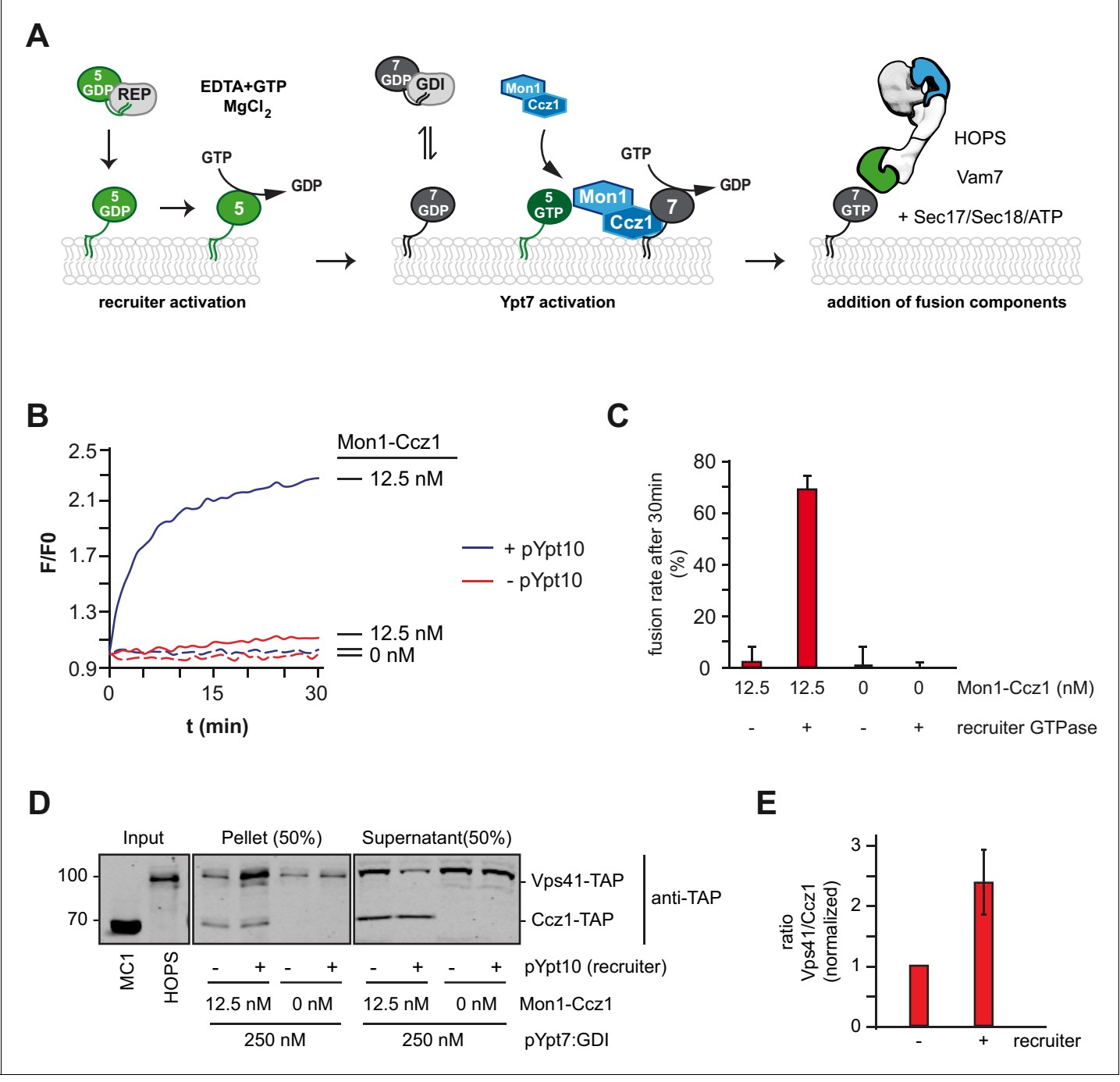

**Figure 5.** Rab5 can trigger Rab7-dependent fusion. (**A**) Scheme of the Rab5-dependent fusion assay. Rab5-proteins in complex with the REP protein were chemically activated. Addition of MANT-GDP-loaded Ypt7:GDI was followed by Mon1-Ccz1 addition. After allowing for nucleotide exchange of Ypt7, fusion machinery was added and fusion reaction was finally triggered by addition of Vam7. (**B**) Fusion depends on the recruiter GTPase. SNARE-bearing proteoliposomes were primed either with chemically activated pYpt10 or mock-treated for 10 min at 27°C. 250 nM Ypt7:GDI and 12.5 nM Mon1-Ccz1 was added, and the reaction was incubated for another 15 min, allowing for nucleotide exchange of Ypt7. Fusion reaction was triggered by addition of 50 nM HOPS, 0.6 μM Sec17, 50 nM Sec18 and 20 nM Vam7. Fusion rate was followed by a content mixing assay, where FRET of enclosed fluorophores is followed (see Methods). (**C**) Fusion rates of (**B**) as measured after 30 min. Error bars represent standard deviation. (**D**) Membrane association of Mon1-Ccz1 and HOPS in fusion assay. Samples of fusion experiments as in (**B**) were recovered after 30 min of measurement. Membrane-bound and soluble fraction were separated by centrifugation for 20 min at 20,000 $g$, and analyzed by SDS-PAGE and Western-Blot. Proteins were detected by using an antibody against the TAP-tag on Vps41 and Ccz1 in the HOPS- and GEF-complex, respectively. (**E**) Densiometric analysis of Western-Blot signals of Vps41 and Ccz1 as shown in (**D**). The ratio of Vps41 over Ccz1-signal is shown for fusion reactions in the presence or absence of a recruiter-GTPase. Signals have been normalized to the Ccz1 signal in the pellet fraction. Error bars represent standard deviation.

*Figure 5 continued on next page*

*Figure 5 continued*

The online version of this article includes the following source data and figure supplement(s) for figure 5:

**Source data 1.** Quantification *Figure 5C* and *Figure 5—figure supplement 1B*.
**Source data 2.** Quantification *Figure 5E*.
**Figure supplement 1.** Fusion of reconstituted proteoliposomes in dependence of Mon1-Ccz1.

---

HOPS was strongly enriched on membranes in the presence of Ypt10 (*Figure 5D,E*). This indicates that Mon1-Ccz1 associates with membranes independently of a recruiter GTPase. Importantly, only in the presence of a Rab5-recruiter, Mon1-Ccz1 can exert sufficient GEF-activity to drive Ypt7 activation, and therey HOPS recruitment and fusion. This suggests that Rab5 directly stimulates catalytic activity of Mon1-Ccz1 beyond promoting binding to the membrane.

## Interaction of Rab5 and Mon1-Ccz1 is regulated by phosphorylation

In a previous study, we identified Mon1 as a target of the vacuolar type 1 casein kinase Yck3 (*Lawrence et al., 2014*). Phosphorylation of Mon1 results in an upshift on SDS gels and release of the protein from vacuoles during in vitro vacuole fusion. Yck3 has several targets at the vacuole, including the HOPS subunit Vps41 and the SNARE Vam3 (*LaGrassa and Ungermann, 2005*; *Brett et al., 2008*). To avoid side-effects of Yck3 on the fusion assay, we turned back to our recruiter GEF assay (*Figure 3*) to ask if Yck3-mediated phosphorylation would affect Mon1-Ccz1 GEF activity. We incubated purified Mon1-Ccz1 with recombinantly produced Yck3 in the presence or absence of ATP. Successful phosphorylation was judged by an upshift of Mon1 on SDS gels (*Figure 6—figure supplement 1*). Using pretreated Mon1-Ccz1, we observed a strong decrease in GEF-activity of the phosphorylated complex (complete) compared to the mock-treated complex, where either Yck3 (no Yck3) or ATP (no ATP) was omitted (*Figure 6A,B*). This shows that Yck3-mediated phosphorylation strongly inhibits Rab5-dependent Mon1-Ccz1 activation.

To control for this observation, we determined the membrane association of the recruiter GTPases pVps21 or pYpt10 after the GEF assay, but did not see any alteration between samples (*Figure 6C*). We also analyzed Mon1 in the same membrane fractions. Phosphorylated Mon1 was less efficiently recruited onto membranes than non-phosphorylated Mon1 in the presence of Vps21 or Ypt10 (*Figure 6C*, lane 1 and 9). We noticed that phosphorylated Mon1 also associated with membrane in the absence of the recruiter GTPase (*Figure 6C*, lane 7 and 15) as observed for non-phosphorylated Mon1 in the fusion assay (*Figure 5D*), yet this association does not have any effect on GEF activity (*Figure 6A*).

We finally predicted that Yck3-mediated phosphorylation could directly affect the productive binding of Mon1-Ccz1 to Rab5-like proteins. To test this, we repeated the interaction analysis of GST-tagged Rab5 homologs Vps21 and Ypt10 with purified Mon1-Ccz1 (*Figure 1D*). In the absence of the kinase Yck3, Mon1-Ccz1 interacted robustly with Ypt10-GTP, and weakly with Vps21-GTP as observed before (*Figure 1D*, *Figure 6D*). The nucleotide specificity of this interaction was completely lost after Yck3-mediated phosphorylation of Mon1-Ccz1 (*Figure 6D*), indicating that phosphorylation blocks the specific interaction with Rab5-proteins and could support Mon1-Ccz1 recycling from membranes (*Figure 6E*).

## Discussion

Understanding the function and regulation of the heterodimeric GEF-complex Mon1-Ccz1 is key to understand the Rab5 to Rab7 transition during endosomal maturation. It was known that Mon1-Ccz1 interacts with Rab5 (*Kinchen and Ravichandran, 2010*; *Cui et al., 2014*) and PI-3-P (*Cabrera et al., 2014*; *Lawrence et al., 2014*; *Hegedűs et al., 2016*). Also, the sequential appearance of Rab5, Mon1, and Rab7 on endosomal membranes was observed in vivo (*Rink et al., 2005*; *Poteryaev et al., 2010*) and has been supported by theoretical models (*Del Conte-Zerial et al., 2008*). In addition, it was shown that the *C. elegans* homologue of Mon1, Sand1, can displace the Rab5 GEF from membranes (*Poteryaev et al., 2010*), which would interrupt the previously described positive-feedback-loop of Rab5 (*Horiuchi et al., 1997*). Despite the fact that the mechanism of Mon1-Ccz1 GEF-activity towards Rab7 has been revealed (*Nordmann et al., 2010*;

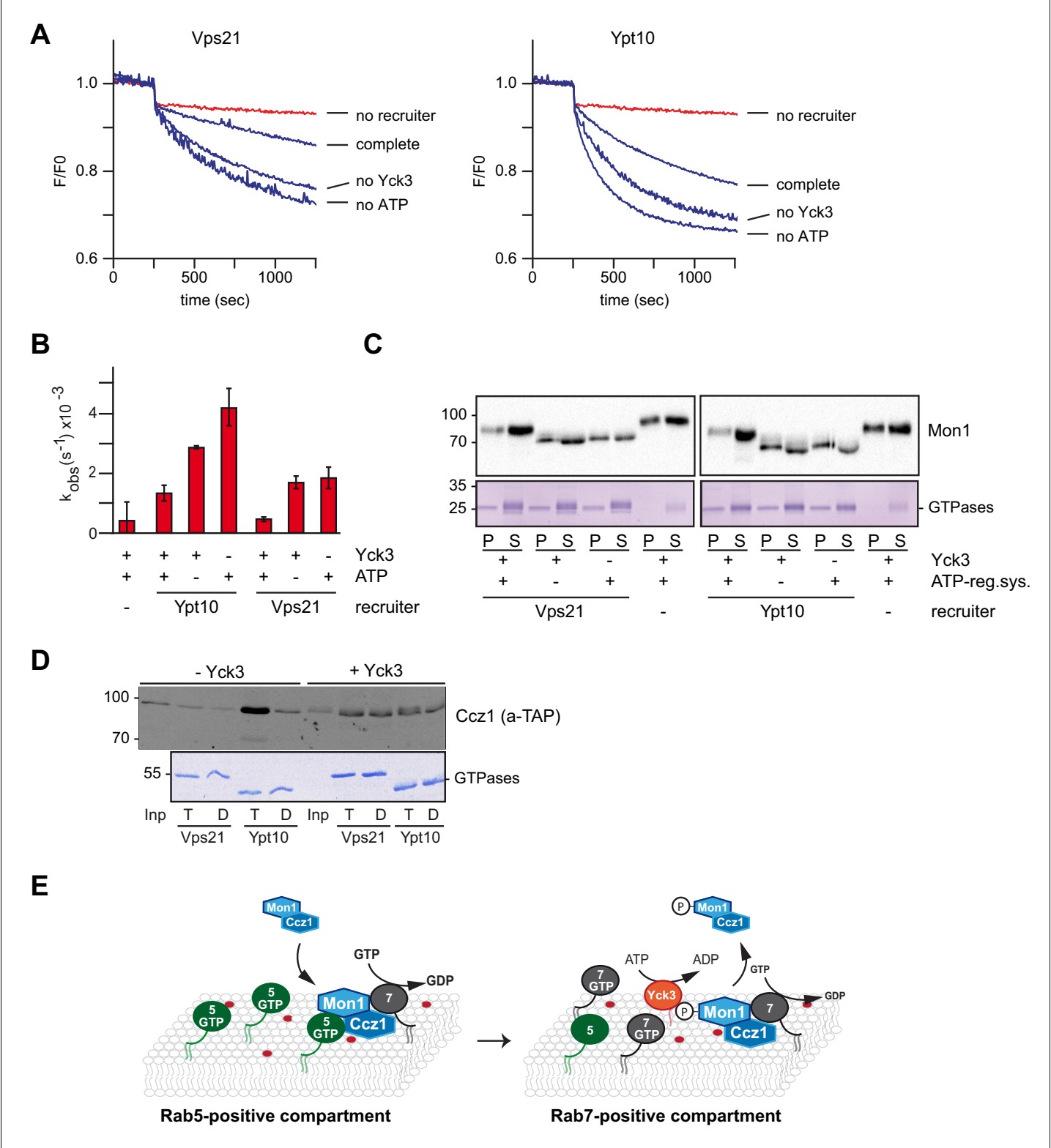

**Figure 6.** The casein kinase Yck3 regulates interaction of Rab5 with Mon1-Ccz1 in yeast. (**A**) Recruiter GEF-assay of phosphorylated Mon1-Ccz1. 50 nM Mon1-Ccz1 were pretreated either with Yck3 and ATP (complete) or with either one of them (no Yck3/no ATP) before it was used in the recruiter GEF assay as described in *Figure 3A*. 1.5 μM GTP-loaded pVps21 or pYpt10 were used as a recruiter GTPase, as control proteoliposomes were mock treated (no recruiter). (**B**) Rate constants as calculated from (**A**). Error bars represent standard deviation. (**C**) Soluble and membrane fraction of recruiter assays in (**A**) were recovered and separated by centrifugation for 20 min at 20,000 *g*, 4°C. Samples were analyzed by SDS-PAGE and Western-Blot for

*Figure 6 continued on next page*

*Figure 6 continued*

presence of Mon1, and by Coomassie staining for used Rab-GTPases. (D) Interaction of yeast Rab5-like proteins with Mon1-Ccz1. Purified GST-tagged Rab5 proteins (Vps21, Ypt10) were loaded with GTP (T) or GDP (D) and incubated with purified Yck3 or mock-treated Mon1-Ccz1 complex. Eluates were analyzed on SDS-PAGE by Western blotting with an antibody against the TAP-tag on Ccz1 (top) and by Coomassie staining (bottom). For details see methods. (E) Working model. Mon1-Ccz1 is recruited to charged membranes. On a Rab5-positive compartment, where Mon1-Ccz1 interacts with Rab5, its GEF-activity towards Rab7 is stimulated. This would lead to a fast transition from a Rab5 to a Rab7-positive compartment. The Rab7-positive compartment converts from an endosomal to a lysosomal/vacuolar membrane. After fusion with the lysosome-like vacuole in yeast, Mon1-Ccz1 is phosphorylated by Yck3. This abolishes the interaction with Rab5 and may in addition lead to a release of Mon1-Ccz1 from the vacuole (*Lawrence et al., 2014*).

The online version of this article includes the following figure supplement(s) for figure 6:

**Figure supplement 1.** Phosphorylation of yeast Mon1-Cz1 by Yck3 in vitro.

---

*Cabrera et al., 2014*; *Kiontke et al., 2017*), there was little mechanistic understanding of how Mon1-Ccz1 function on endosomes is coordinated in vivo.

Here we demonstrate the first reconstitution of the endosomal Rab-cascade, where we recapitulate the transition from Rab5 to Rab7. We show that the Mon1-Ccz1 GEF complex has a dual membrane targeting mechanism by binding both Rab5 and charged lipids like PI-3-P. Phospholipid interactions show no specificity for particular head groups but are driven by electrostatic interactions. Our data reveal that Mon1-Ccz1 is an evolutionarily conserved effector of Rab5, both in yeast and *Drosophila*. Once bound to Rab5-GTP, it can activate and thereby stably localize Rab7, which was previously soluble and in complex with GDI. Using the yeast system, we show that Rab5-GTP can be limiting for Ypt7 and HOPS driven, SNARE-dependent membrane fusion. Importantly, Rab5 binding strongly stimulates the catalytic activity of Mon1-Ccz1. The process of Mon1-Ccz1 membrane association and Rab5 interaction can be regulated by casein kinase-mediated Mon1-Ccz1 phosphorylation, most likely by promoting release of Mon1-Ccz1 from Rab5 and thus vacuoles (*Figure 6E*).

While our data now recapitulate the essential role of Mon1-Ccz1 in Rab7 recruitment and activation from the GDI complex, we do not yet understand the exact crosstalk of Mon1-Ccz1 with Rab5 and PI-3-P levels on endosomes. The PI-3-Kinase Vps34 is a Rab5 effector (*Christoforidis et al., 1999*), and thus Rab5 and PI-3-P levels should both increase when endosomes grow and mature. Endosomal Rabs are not uniformly distributed along the endosomal membrane (*Sönnichsen et al., 2000*; *Franke et al., 2019*), which could result in selective enrichment of Mon1-Ccz1 and thus Rab7 to distinct domains on endosomes. Moreover, in vivo analyses showed the transition from Rab5 to Mon1 and Rab7 (*Poteryaev et al., 2010*). Our data suggest that the interaction with charged lipids is insufficient for Mon1-Ccz1 GEF activity, yet may help for initial targeting to endosomes or autophagosomes (*Hegedűs et al., 2016*). In agreement, we observe a synergistic effect of charged lipids and Rab5 in Mon1-Ccz1 binding to membranes (*Figure 2*). However, for GEF-activity of Mon1-Ccz1 towards prenylated Rab7/Ypt7:GDI recruitment by charged lipids was insufficient, and we required membrane-bound Rab5-GTP in the recruiter GEF assay, suggesting that Rab5 has a critical, and possibly rate-limiting role in Mon1-Ccz1 activity on membranes.

By utilizing the *Drosophila* Rab5-Rab7 system, we now provide evidence that the Rab5-dependent Mon1-Ccz1 GEF-activity is also conserved across species, and likely extends to the homologous mammalian complex (*Vaites et al., 2018*). We could not observe any strong difference in the behavior of the dimeric and trimeric Rab7-GEF-complex, indicating that the third subunit does not influence the overall GEF-activity as suggested (*Vaites et al., 2018*). Our data thus agree with our structural analyses of the Mon1-Ccz1 core complex, where we showed that the GEF activity depends solely on the Mon1-Ccz1 interface (*Kiontke et al., 2017*). Likewise, human dimeric Mon1-Ccz1 complex is sufficient to activate Rab7 (*Gerondopoulos et al., 2012*), in agreement with recent findings on the related Rab23 GEF complex of Inturned-Fuzzy and its third subunit WDECP/fritz (*Gerondopoulos et al., 2019*). Moreover, our data directly show that Rab5-interaction with the Mon1-Ccz1 subcomplex of the trimeric complex is preserved across species and does not depend on the third subunit. It is likely that the third subunit acts beyond the Mon1-Ccz1 GEF activity, an issue for future analyses.

In the yeast system, we identify the Rab5-like Ypt10 as a new interactor of Mon1-Ccz1. Ypt10 colocalizes with and strongly binds to Mon1-Ccz1 (*Figure 1D,E*). We thus consider Ypt10 as a

member of the Rab5-family in yeast. In agreement, overproduction of Ypt10 affects the endomembrane system and cell growth (*Louvet et al., 1999*), even though its detailed function on endosomal membranes remains to be analyzed.

In sum, our findings show that the GEF Mon1-Ccz1 is both necessary and sufficient for the endosomal Rab5 to Rab7 transition, in agreement with in vivo analyses (*Poteryaev et al., 2010*). They thus explain, how GEF localization and activation can drive organelle transition, and how Rabs can thus act consecutively. This model nicely agrees with previously implied Rab cascades, which were mainly based on in vivo correlation in Rab and GEF localization and protein-protein interactions (*Poteryaev et al., 2010*; *Hutagalung and Novick, 2011*; *Pusapati et al., 2012*; *Barr, 2013*; *Pei et al., 2014*; *Stalder and Novick, 2016*; *Yamano et al., 2018*). Importantly, we now directly demonstrate the consecutive Rab activation by reconstituting the Rab5 to Rab7 cascade with purified components. Intriguingly, not only Rabs, but also an Arf GTPase can function as a recruiter GTPase (*Thomas and Fromme, 2016*), which shows that GEFs use several organelle-specific small GTPases along the endomembrane system.

We propose a working model, in which Mon1-Ccz1 is recruited by charged lipids such as PI-3-P, and Rab5 to the endosomal membrane. Rab5 also directly stimulates Mon1-Ccz1, resulting in recruitment and activation of Rab7 out of the GDI complex (*Figure 6E*). Mon1-mediated displacement of the Rab5 GEF (*Poteryaev et al., 2010*) would then favor the transition of a Rab5 to a Rab7-positive endosome, which subsequently can fuse with the lysosome-like vacuole in yeast (*Langemeyer et al., 2018a*). Here, the vacuolar casein kinase Yck3 phosphorylates Mon1-Ccz1 and thus disrupts the interaction with Rab5. We realize that this observation has similarity to the function of the homologous Yck1 and 2 casein kinases in the late secretory pathway in yeast. These kinases phosphorylate the GEF Sec2 and thus promote an interaction with Sec15, an effector of the Rab GTPase Sec4 and subunit of the exocyst tethering complex, which results in a positive-feedback loop (*Stalder and Novick, 2016*). Once the Mon1-Ccz1 binding is lost, Rab5 can interact with its GAP and become a substrate of GDI-mediated extraction. Remaining phospholipid-bound Mon1-Ccz1 will activate Rab7 at a lower rate, maintaining the Rab7-domain on late endosomes or the vacuole (*Figure 6E*). Alternatively, the complex may be released from vacuoles due to changes in the lipid composition. In agreement, Mon1-Ccz1 is found almost exclusively on endosomal dots proximal to the vacuole in yeast cells (*Figure 1*). Future studies will need to clarify how Rab7 activation is coordinated with Rab5 inactivation to understand the entire cascade.

## Materials and methods

### Strains and plasmids

Strains used in this study are listed in *Supplementary file 1*. Deletions and tagging of genes were done by PCR-based homologous recombination with appropriate primers (*Puig et al., 1998*; *Janke et al., 2004*). The plasmids used are listed in *Supplementary file 2*. A CRISPR-Cas9 approach was selected to generate endogenously GFP-tagged Ypt10 (*Generoso et al., 2016*).

### Fluorescence microscopy

Cells were grown to logarithmic phase in synthetic medium, supplemented with essential amino acids (SDC). The vacuolar membrane was stained by addition of 30 µM FM4-64 for 30 min, followed by washing and incubation in medium without dye for 1 hr as described (*Vida and Emr, 1995*). Cells were imaged on an Olympus IX-71 inverted microscope equipped with 100x NA 1.49 and 60x NA 1.40 objectives, a sCMOS camera (PCO, Kelheim, Germany), an InsightSSI illumination system, 4′,6-diamidino-2-phenylindole, GFP, mCherry, and Cy5 filters, and SoftWoRx software (Applied Precision, Issaquah, WA). We used z-stacks with 0.4 µM. All microscopy image processing and quantification was performed using ImageJ (National Institutes of Health, Bethesda, MD).

### Expression and purification of Rab GTPases

GST-TEV-Ypt7, GST-TEV-Vps21, GST-TEV-Rab5 and GST-TEV-Rab7 were expressed and purified with slight modifications as described (*Nordmann et al., 2010*). Shortly, cells were lysed in a Microfluidizer, Model M-110L (Microfluidics, Newton, MA), the lysate was clarified by centrifugation at 40,000 $g$ for 30 min at 4°C, and loaded onto a pre-equilibrated Protino GST/4B 1 ml column

(Macherey and Nagel, Germany). The column was extensively washed and eluted in lysis buffer (50 mM Tris, pH 7.4, 300 mM NaCl, 2 mM MgCl$_2$) containing either 20 mM Glutathione or TEV-protease for cleavage at 4°C overnight. Proteins were afterwards dialyzed into assay buffer (50 mM HEPES, NaOH pH 7.4, 150 mM NaCl, 1 mM MgCl$_2$), changing the buffer twice.

## Expression and Purification of Rab GGTase and Rab Escort Protein

pGATEV-Bet2 and pET30a-Bet4 (*Kalinin et al., 2001*) were co-expressed in *E. coli* BL21 Rosetta, induced with 0.25 mM IPTG for 16 hr at 18°C. The Rab Escort Protein Mrs6 (*Pylypenko et al., 2003*) was expressed and purified the same way. Cells were lysed in 50 mM Tris, pH 8.0, 300 mM NaCl, and 2 mM beta-mercaptoethanol, 1 mM PMSF as described above. The cleared lysate was loaded on a Hi-Trap Ni-Sepharose column (GE, Germany) equilibrated with the lysis buffer. The column was washed extensively with lysis buffer containing 30 mM imidazole. Bound protein was eluted with a linear 30 to 300 mM imidazole gradient over 30 column volumes. Fractions containing GGTase-II and REP, respectively, were pooled and dialyzed against assay buffer, which was changed twice.

## Expression and purification of Gdi1

Competent *E. coli* Rosetta BL21 cells were transformed with pGEX-6P-Gdi1 (*Thomas and Fromme, 2016*) or pET28a-His-SUMO-d.m.GDI. A single colony was picked from selection plates, and 2 l of culture was grown to an OD$_{600}$ of around 0.8. Expression was induced with 0.25 mM IPTG, and cells were incubated at 18°C overnight. Cells were lysed in PBS containing 1 mM PMSF, 2 mM MgCl$_2$, 5 mM beta-mercaptoethanol, and the cell homogenate was cleared as described above. Cleared lysate was loaded on a Protino GST/4B 1 ml column (Macherey and Nagel, Germany) or a Protino Ni-NTA 1 ml column (Macherey and Nagel, Germany). Protein was eluted after extensive washing by cleaving the affinity-tag incubating with Precision- or SUMO-protease at 16°C for 2 hr or at 4°C overnight, respectively.

## Expression and purification of *Drosophila* GEF complexes from Sf21 cells

Sf21 cells developed from ovary tissue of Spodoptera frugiperda were grown as a monolayer culture in Insect-XPRESS Protein-free Insect Cell Medium (Lonza) at 27°C and standard T175 culture flasks. GEF subunits were expressed using the biGBac system with FuGENE6 as transfection reagent (Promega) (*Weissmann et al., 2016*). For large scale protein purification, cells were infected with recombinant viruses encoding Mon1-Ccz1 or Mon1-Ccz1-CG8270 for 72 hr. Cells were harvested by centrifugation for 5 min at 500 *g* and stored at −80°C until usage. Cells were lysed in buffer containing 50 mM HEPES-NaOH pH 7.5, 300 mM NaCl, 10% glycerol 1 mM phenylmethylsulfonyl fluoride and 1x protease inhibitor cocktail using the microfluidizer (Microfluidics). Lysates were centrifuged for 30 min at 40,000 *g*, and the cleared lysates were incubated with 1 ml Glutathione sepharose 4B (GE Healthcare) equilibrated with 50 mM HEPES-NaOH pH 7.5, 300 mM NaCl and 10% glycerol for 2 hr at 4°C on a nutator. Beads were transferred to Mobicol-columns (MoBiTec GmbH), and each Mobicol was washed with 15 ml buffer. Elution of the complexes was performed by incubation with PreScission protease (0.4 mg/ml) in the presence of 1 mM dithiothreitol overnight at 4°C on a turning wheel. Complexes were eluted from the Mobicols by centrifugation, and the beads were washed twice with buffer. Elutions were combined and concentrated to a volume of 500 µl using a Vivaspin 6 10,000 MWCO centrifugal concentrator (Sartorius). The concentrated sample was then subjected to size exclusion chromatography using an Äkta FPLC UPC-900 liquid chromatography system (GE Healthcare) equipped with a Superdex 200 increase 10/300 GL column (GE Healthcare) against a buffer containing 50 mM HEPES-NaOH pH 7.5, 300 mM NaCl, 1 mM MgCl$_2$ and 10% glycerol. Peak fractions were collected and analyzed by SDS-PAGE.

## Tandem Affinity Purification

Purifications were conducted essentially as described (*Ostrowicz et al., 2010*; *Bröcker et al., 2012*). Three liters of culture were grown at 30°C to OD$_{600}$ of 5. Cells were harvested by centrifugation and lysed in buffer containing 50 mM HEPES-NaOH, pH 7.4, 150 mM NaCl, 1.5 mM MgCl$_2$, 1xFY protease inhibitor mix (Serva, Germany), 0.5 mM PMSF and 1 mM DTT. Lysates were centrifuged for 90 min at 100,000 *g*, and supernatants were incubated with IgG Sepharose (GE, Germany) for 1.5 hr at

4°C. Beads were sedimented by centrifugation at 800 $g$ for 5 min, and washed with 15 ml lysis buffer containing 0.5 mM DTT. Bound proteins were eluted by TEV cleavage overnight, and analyzed on SDS-PAGE.

## Rab pull down

Recombinant GST-fusion proteins (75 µg per sample) were incubated with 500 µl of 20 mM HEPES-NaOH, pH 7.4, 20 mM EDTA and 10 mM GDP or GTP (Sigma Aldrich, Germany). After incubation for 30 min at 30°C, samples were adjusted to 25 mM $MgCl_2$ and 7 mg/ml bovine serum albumin, loaded onto 30 µl prewashed GSH-Sepharose 4B (GE Healthcare), and incubated for 1 hr at 4°C. After incubation, GSH-Sepharose was spun for 1 min at 2000 $g$, and the supernatant was discarded. The pellet was resuspended in 200 µl of pulldown buffer containing 20 mM HEPES-NaOH, pH 7.4, 150 mM NaCl, 1 mM $MgCl_2$, 5% (v/v) Glycerol and 0.1% (v/v) Triton X-100. Suspension was then incubated with 5 µl of the corresponding nucleotide (100 mM stock solution), 50 µl of a 70 mg/ml BSA solution and 50 µg Mon1-Ccz1 for 1 hr at 4°C on a turning wheel. Beads were washed three times with 500 µl pull down buffer. Bound proteins were finally eluted by addition of 500 µl elution buffer (20 mM HEPES-NaOH, pH 7.4, 150 mM NaCl, 20 mM EDTA, 5% (v/v) Glycerol and 0.1% (v/v) Triton X-100 for 20 min at room temperature on a turning wheel. Elution fraction was TCA-precipitated and analyzed by SDS-PAGE and Western-Blot. In all shown gels, 0.5% of the input fraction and 20% of the elution fraction was loaded. To compare amounts of immobilized Rab-GTPases, 1x Laemmli buffer was added to GSH-Sepharose, and samples were boiled for 10 min at 95°C. 1% of this bead sample was analyzed by SDS-PAGE and Coomassie staining.

## In vitro prenylation of Rab GTPases

To obtain prenylated Rab-chaperone complexes the prenylation reaction was performed as described previously (*Langemeyer et al., 2018b*; *Thomas and Fromme, 2016*). Rab-GTPases were preloaded with either GDP (Sigma Aldrich, Germany) or MANT-GDP (Jena Bioscience, Germany). For *Drosophila* or yeast Rab-GTPases in complex with GDI, the respective purified GDI from the same organism was used. To complex Rab-GTPases with REP, the yeast REP was used.

## Nucleotide exchange assays

Recruiter GTPase dependent Guanine Nucleotide Exchange Factor (GEF) assays were performed by preincubating 333 µM liposomes composed of a vacuolar mimicking lipid mix (VML; 47.35 or 46.1 mol % dioleoyl phosphatidylcholine [DOPC], 18% dioleoyl phosphatidylethanolamine [DOPE], 1% diacylglycerol, 8% ergosterol, 2% dioleoyl phosphatidic acid [PA], 18% soy phosphatidylinositol [PI], 4.4% dioleoyl phosphatidylserine [DOPS], 1% dipalmitoyl PI-3-phosphate [PI-3-P] (Life Technologies) *Zick and Wickner, 2014*) with 1.5 µM prenylated recruiter GTPase complexed with REP and 200 µM GTP in the presence of 1.5 mM EDTA for 10 min at 27°C. Loading reaction was stopped by addition of 3 mM $MgCl_2$. 250 nM MANT-GDP loaded pYpt7:GDI was added to the preincubated liposomes, and volume was adjusted to 800 µl and transferred to a fluorimeter with a temperature-controlled cuvette and a stirring device (Jasco, Gross-Umstadt, Germany). The MANT-fluorophore was excited at 355 nm, and emission was recorded over time at 488 nm at 27°C. After allowing for baseline stabilization, indicated amounts of Mon1-Ccz1 were added to trigger the nucleotide exchange reaction of Ypt7. Release of MANT-GDP from the binding pocket resulted in a quench of fluorescence. Data were fitted against a first-order exponential decay and $k_{obs}$ was determined. Subsequently, $k_{obs}$ was plotted against the GEF-concentration and $k_{cat}/K_M$ ($M^{-1}s^{-1}$) was determined as the slope of the linear fit.

## Phosphorylation of Mon1-Ccz1

Purified Mon1-Ccz1 was incubated with a 10x molar excess of recombinantly produced Yck3 in the presence of ATP regenerating system (0.5 mM ATP, 0.1 mg/ml creatine kinase, 40 mM creatine phosphate, 1 mM PIPES, pH 6.8, 20 mM Sorbitol) for 30 min at 27°C. Successful phosphorylation was judged by analyzing samples by SDS-PAGE and Western-Blot. Phosphorylation resulted in an upshift of the band of Mon1 on gels.

## Fusion assay with reconstituted proteoliposomes (RPLs)

Fusion of RPLs, including the purification of the involved proteins, was performed as described (*Langemeyer et al., 2018b*) with a protein to lipid ratio of 1:10,000. RPLs were composed of the VML-lipid mix, as described above, and either 0.25% Marina Blue-PE or 1.5% nitrobenzoxadiazole (NBD)-PE (*Zick and Wickner, 2014*). One set of RPL carried Nyv1, the other set Vti1 and Vam3. RPLs were preincubated with prenylated recruiter Rab-GTPase, as indicated. The mixture was incubated for 10 min at 27°C to allow for membrane association of the recruiter Rab-GTPase and nucleotide exchange in the presence of 0.5 mM GTP and 3 mM EDTA. The reaction was stopped by addition of 5 mM $MgCl_2$. Afterwards 250 nM pYpt7:GDI and indicated amounts of Mon1-Ccz1 were added. The reaction was transferred to a SpectraMax M3 Multi-Mode Microplate Reader (Molecular Devices, Germany), and incubated for 15 min at 27°C to allow for nucleotide exchange while recording the fluorescence signals. Fusion was triggered by addition of 50 nM HOPS complex, 50 nM Sec18, 600 nM Sec17, and finally 100 nM Vam7. To follow full fusion of liposomes, content mixing and subsequent increase in fluorescence was monitored.

## Fractionation of recruiter GEF-assay and RPL-fusion assay

To test for membrane association in the recruiter GEF-assay and the RPL-fusion assay, the reactions were followed in a fluorimeter and SpectraMax M3 Multi-Mode Microplate Reader (Molecular Devices, Germany) for 30 min. Afterwards, samples were transferred to centrifugation tubes and spun for 20 min, 20,000 *g* at 4°C. Pellet and supernatant fractions were separated. The volume was adjusted to 500 µl with assay-buffer containing 0.1% Triton-X 100. Proteins were precipitated by addition of 13% trichloro acetic acid (TCA) followed by wash with 100% ice-cold acetone, and analyzed by SDS-PAGE and Western-Blot. Proteins were detected using antibodies against GST-Ypt7 and the TAP Tag antibody (Invitrogen, CAB1001) directed against the TAP-tag at Vps41 and Ccz1, respectively.

## Liposome Sedimentation Assay

Liposomes were generated from neutral (82 mol% 1-palmitoyl-2-oleoyl PC [POPC], and 18 mol% 1-palmitoyl-2-oleoyl PE [POPE]) or PIP containing (79 mol% POPC, 18 mol% POPE, 2 mol% PI-3-P, and 1 mol% dipalmitoylphosphatidylinositol-3–5-bisphosphate [PI-3,5-P$_2$]) lipid mixtures. The lipids were dried for at least 1 hr in a speedvac, and dissolved in 1 ml of sedimentation buffer (25 mM HEPES, pH 7.3, 250 mM NaCl, 1 mM $MgCl_2$ and 5% sucrose) to a final lipid concentration of 2 mM. The liposome suspension was freeze/thawed for 10 times in liquid nitrogen and at 56°C respectively. To co-pellet the protein and the liposomes, final concentrations of 0.5 µM lipids, 1 µM prenylated Vps21, 25 µM GTP, and 1 µM yeast Mon1-Ccz1 were added in buffer without sucrose to a final volume of 200 µl. After 20 min of incubation at room temperature, the liposomes were pelleted at 20,000 *g* for 20 min at 4°C. The soluble fraction in the supernatant was separated from the membrane fraction in the pellet and precipitated with ice-cold 100% acetone, while the pellet was kept on ice. All samples were analyzed via SDS-PAGE and subsequent Coomassie staining. The gels were scanned and the band intensity was quantified with Biorad Image Lab. Significance analysis were performed by two-tailed heteroscedastic t-test statistics (*, $p \leq 0.05$).

## Acknowledgements

We thank Kumar Verma for help with the initial setup of the *Drosophila* system, Claudine Kraft for helpful support with protein purifications, and Maya Schuldiner, Kirill Alexandrov and Roger Goody for kindly providing reagents. This work was supported by the SFB 944 (project P11, to CU, and project P17, to DK).

# Additional information

## Funding

| Funder | Grant reference number | Author |
| --- | --- | --- |
| Deutsche Forschungsgemeinschaft | SFB 944 | Daniel Kümmel Christian Ungermann |

The funders had no role in study design, data collection and interpretation, or the decision to submit the work for publication.

#### Author contributions
Lars Langemeyer, Conceptualization, Data curation, Formal analysis, Validation, Investigation, Visualization, Methodology, Writing - original draft, Writing - review and editing; Ann-Christin Borchers, Data curation, Formal analysis, Validation, Investigation, Visualization, Methodology, Writing - review and editing; Eric Herrmann, Data curation, Formal analysis, Validation, Visualization, Methodology; Nadia Füllbrunn, Data curation, Formal analysis, Validation, Investigation, Methodology; Yaping Han, Angela Perz, Formal analysis, Validation, Investigation, Methodology; Kathrin Auffarth, Formal analysis, Validation, Visualization, Methodology; Daniel Kümmel, Conceptualization, Formal analysis, Supervision, Funding acquisition, Validation, Methodology, Project administration, Writing - review and editing; Christian Ungermann, Conceptualization, Funding acquisition, Validation, Writing - original draft, Project administration, Writing - review and editing

#### Author ORCIDs
Lars Langemeyer (ID) https://orcid.org/0000-0002-4309-0910
Daniel Kümmel (ID) http://orcid.org/0000-0003-3950-5914
Christian Ungermann (ID) https://orcid.org/0000-0003-4331-8695

#### Decision letter and Author response
Decision letter https://doi.org/10.7554/eLife.56090.sa1
Author response https://doi.org/10.7554/eLife.56090.sa2

## Additional files

#### Supplementary files
• Source data 1. Enzyme kinetics data of *Figure 3D* and *Figure 6B*.

• Supplementary file 1. Strains used in this study.

• Supplementary file 2. Plasmids used in this study.

• Transparent reporting form

#### Data availability
All data generated or analysed during this study are included in the manuscript and supporting files.

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
