## [Decision Letter]

**Acceptance summary:**

Rab GTPases control the passage of membrane proteins through the secretory and endocytic pathways and regulate each other in an ordered, cascade fashion to provide directionality for these pathways. Here, the authors set out to define the minimal molecular determinants for conversion of Rab5 positive endosomes to Rab7 positive endosomes, specifically how Rab5 promotes Rab7 activation through the Mon1-Ccz1 GEF complex and how Rab5 interaction with the GEF is modulated by casein kinase. This elegant biochemical study provides important molecular detail to a key step in progression along the endocytic pathway.

**Decision letter after peer review:**

Thank you for submitting your article "A conserved and regulated mechanism drives endosomal Rab transition" for consideration by *eLife*. Your article has been reviewed by three peer reviewers, and the evaluation has been overseen by Suzanne Pfeffer as the Senior and Reviewing Editor. The following individuals involved in review of your submission have agreed to reveal their identity: Marino Zerial, Chris Fromme and Francis Barr.

The reviewers have discussed the reviews with one another and the Reviewing Editor has drafted this decision to help you prepare a revised submission.

Ungermann and co-workers set out to define the minimal molecular determinants for conversion of Rab5 positive endosomes to Rab7 positive endosomes, a process that occurs in cells during endosome to lysosome maturation. To fully explain this process, we need to understand how Rab5 promotes Rab7 activation, and how Rab7 promotes Rab5 release. The focus of this work is very much on how Rab5 promotes Rab7 activation through the Mon1-Ccz1 GEF complex and how Rab5 interaction with the GEF is modulated by casein kinase. The authors provide a compelling case addressing this first part, and how the Rab5-dependent pool of Mon1-Ccz1 is finally released by casein kinase activity.

The reviewers were all very positive about this story and would like to see it in *eLife* after some additional issues regarding the physiological relevance of Ypt10 and its relationship to Ypt51/52/53 are addressed. I have included all the reviewer comments as many are textual and come from the leaders in this field so will benefit your efforts during revision. It is not necessary to do all the requested experiments but please try to address these issues as best as possible.

Reviewer #1:

This manuscript delves into the Rab5 to Rab7 conversion process in a thoughtful, well-executed and well-written manner, besides some incorrect choice of references. I would recommend publication. Their stated goals of determining whether Mon1-Ccz1 is sufficient to drive conversion and whether both active Rab5 and PI3P are necessary for the conversion process are satisfactorily addressed with very interesting results.

One intriguing conclusion is "Importantly, only in the presence of a Rab5-recruiter, Mon1-Ccz1 can exert sufficient GEF-activity to drive Ypt7 activation, and thereby HOPS recruitment and fusion." If one thinks more broadly (including considering mammalian cells), it follows that Mon1-Ccz1 could not activate Rab7 once Rab5 has been removed by conversion. Ypt10 and Rab5 are quite different (primary sequence), can Ypt10 really be considered equivalent to Rab5? The authors should clarify how this would work in vivo.

One of the more intriguing concepts was the possible regulation by a third subunit, creating a trimeric GEF complex in metazoans and *Drosophila*. Given the suggestion by Pontano-Vaites et al. that mammalian RMC1 regulates Mon1-Ccz1, it was (mildly) disappointing that the trimeric *Drosophila* Bulli-Mon1-Ccz1 complex displayed GEF activity similar to that of the dimeric Mon1-Ccz1 complex. The authors cite "Dehnen et al., submitted" for a description of the *Drosophila* system. However, this topic is sufficiently interesting that having access to data in the Dehnen et al. manuscript is crucial. For example, are the sequences of RMC1 and Bulli similar or has the Mon1-Ccz1 complex evolved two alternative pathways for regulation? I do not mean to ask that data of Dehnen et al. manuscript are incorporated in this study, but it should be made available, e.g. in bioRxiv.

Another aspect of the work which was fascinating is the regulation of Mon1-Ccz1 by the Rab protein Ypt10p. Based on the fact that Ypt10p binds Mon1-Ccz1 with higher affinity than other Rab proteins, one would expect the enhancement of GEF activity to be caused by more Mon1-Ccz1 bound to the liposome. However, the authors showed this was not the case, suggesting an actual regulatory mechanism for Ypt10. Delving into this mechanism more thoroughly would make this story of interest to a wider audience. For example, can soluble Ypt10 stimulate Mon1-Ccz1 activity? It would be nice if the authors could provide some insights into this mechanism. While I don't think it's mandatory for publication, it is at least a topic for further discussion.

Reviewer #2:

Ungermann and co-workers set out to define the minimal molecular determinants for conversion of Rab5 positive endosomes to Rab7 positive endosomes, a process that occurs in cells during endosome to lysosome maturation. To fully explain this process, we need to understand how Rab5 promotes Rab7 activation, and how Rab7 promotes Rab5 release. The focus of this work is very much on how Rab5 promotes Rab7 activation through the Mon1-Ccz1 GEF complex and how Rab5 interaction with the GEF is modulated by casein kinase. The authors provide a compelling case addressing this first part, and how the Rab5-dependent pool of Mon1-Ccz1 is finally released by casein kinase activity. However, this leaves open the question of how Rab7 feeds back on to Rab5 to promote Rab5 release. Previous literature provides some clues to this, and the authors should adjust their Abstract, manuscript text and model to better reflect this.

I will address the three key questions raised by the authors in their Abstract:

i) That Mon1-Ccz1 is an effector of Rab5.

This claim is based on observations that Ypt10 can interact with Mon1-Ccz1 and related Rabs. Overall the data is convincing. However, the hypothesis was surely that Ypt51/52/53 recruit Mon1-Ccz1, so there is a need to look at all these proteins together. Figure 1E shows Ypt10 and Ccz1, can the authors show similar data for Vps21 and Ypt52 which have a much greater effect on Ypt7 localisation (Figure 1F)?

Figure 1D and Figure 1—figure supplement 1. Can the authors show the other subunits of the GEF, not only Ccz1. This is necessary to conclude that the GEF complex has bound, not only one subunit. How reproducible is this? Graphed data from multiple experiments would address this.

ii) That membrane-bound Rab5 is the key factor to directly promote Mon1-Ccz1 dependent Rab7 activation and Rab7- dependent membrane fusion.

Experiments on the interaction of Mon1-Ccz1 with membranes are done with Vps21 and PIPs (Figure 2), when based on the data in Figure 1 Ypt10 would be more appropriate. This becomes important for Figure 3, when the effects on Rab7 GEF activity are tested. Ypt10 shows a robust effect while the others Rab5-related proteins show a more modest stimulation. This is not reflected in the membrane pelleting shown in Figure 3E, where the amount of Mon1-Ccz1 recovered is equivalent in all cases.

Figure 5 switches back to Ypt10, and shows very clear evidence that it promotes Mon1-Ccz1 GEF activity and membrane fusion. Does this work for the other Rab5 family members?

iii) That this process is regulated in yeast by the casein kinase Yck3, which phosphorylates Mon1 and blocks Rab5 binding. Our study thus uncovers the minimal feed-forward machinery of the endosomal Rab cascade and novel regulatory mechanism, which can explain organelle maturation in eukaryotic cells.

This conclusion is supported by Figure 6 and the data show Yck3 blocks both Vps21 and Ypt10 interaction, presumably by acting at the same site on the complex. Do the authors have any idea where Yck3 phosphorylates the Mon1-Ccz1 complex? It is unlikely but can we completely exclude the possibility that the Rabs are Yck3 targets?

Reviewer #3:

This study from Langemeyer et al. presents an in vitro reconstitution reaction in which yeast Rab5-GTP recruits Mon1/Ccz1, the GEF for Rab7, to liposome membranes. The field has largely coalesced around support for a model of a Rab5-Rab7 cascade on endosomes. However, whether activated Rab5 directly interacts with Mon1/Ccz1 has not been clearly demonstrated. The authors present several convincing pieces of evidence in support for a direct role of activated Rab5 in recruiting Mon1/Ccz1 to endo-lysosomal membranes in order to activate Rab7 (Ypt7).

A summary of their findings:

The authors find that of the four yeast Rab5 paralogs, Ypt10 binds to Mon1/Ccz1 most strongly in vitro. They find that Mon1/Ccz1 prefers anionic membranes in vitro. They find that all four Rab5 paralogs stimulate Mon1/Ccz1 GEF activity towards Ypt7 on liposome membranes. They recapitulate these findings with *Drosophila* Rab5, Mon1/Ccz1, and Rab7 proteins. They find that the *Drosophila* Bulli protein, a third subunit of the complex, is dispensable for Rab5 binding and stimulation of Rab7 GEF activity. They are able to reconstitute a membrane fusion reaction in which Rab5-GTP is required to stimulate Mon1/Ccz1 activation of Ypt7. They show that the Yck3 kinase could potentially regulate this process because its phosphorylation of Mon1 reduces the strength of the in vitro interaction with Rab5-GTP. They perform some in vivo experiments showing that certain double or triple-mutant yeast strains are defective in trafficking to the vacuole or mislocalize Ypt7.

Overall, I find the in vitro experiments to be a compelling demonstration of this Rab cascade. However, I think the in vivo experiments as presented do not currently provide sufficient information needed to properly assess the true physiological relevance of the Ypt10 interaction.

1) The authors find that Ypt10 is the best physical interactor with Mon1/Ccz1, yet they have not thoroughly tested its importance in vivo. As shown in Figure 1F, Ypt10 does not appear to be required for Ypt7 activation in cells, but this could be due to redundancy with the other Rab5 paralogs. The authors should determine whether double mutants of ypt10 and vps21 (and perhaps other double-mutant combinations) perturb Ypt7 activation in cells.

2) I'm also surprised the authors did not examine localization of Mon1/Ccz1 in various Rab5 mutants. Shouldn't there be a Mon1/Ccz1 mislocalization phenotype in some combination of Rab5 homolog (double, triple, etc). mutant cells, especially involving ypt10?

3) Furthermore, how do the authors explain the punctate localization of Ypt7 in vps21-ypt52 double mutant cells? The punctate localization suggests Ypt7 is still being activated in these mutants, and is activated at an endosomal structure rather than on the vacuole. In fact, one interpretation of this observation is that Mon1/Ccz1 localization and function is unperturbed in the vps21-ypt52 double mutants, and instead the maturation process is perturbed.

4) The authors switch which Rab homolog they are examining in different experiments. The authors state that they would use Ypt10 in "further assays", yet in Figure 2 they use Vps21 to test liposome-binding of Mon1/Ccz1. Why wasn't Ypt10 also tested in Figure 2?

5) Similarly, as I was reading the manuscript, in Figure 2 I was expecting a direct comparison of liposome binding of Mon1/Ccz1 for all of the yeast Rab5 paralogs, as well as at least one negative control (i.e. Ypt6 or Ypt31). The experiments shown in Figure 3, in which the specificity of several different Rabs in stimulating Mon1/Ccz1 GEF activity is tested, could potentially obviate the need for such an experiment. However, in the second paragraph of the subsection “Control of Mon1-Ccz1 GEF activity by Rab5 is evolutionarily conserved” the authors claim that the GEF activity correlates with the affinity of the effector interaction between the Rab and Mon1/Ccz1 – I think in order to make this claim they would need to compare the relative strengths of the different Rabs in recruiting Mon1/Ccz1 to membranes through liposome binding experiments similar to those shown in Figure 2.

---

## [Author Response]

Reviewer #1:[…] One intriguing conclusion is "Importantly, only in the presence of a Rab5-recruiter, Mon1-Ccz1 can exert sufficient GEF-activity to drive Ypt7 activation, and thereby HOPS recruitment and fusion." If one thinks more broadly (including considering mammalian cells), it follows that Mon1-Ccz1 could not activate Rab7 once Rab5 has been removed by conversion. Ypt10 and Rab5 are quite different (primary sequence), can Ypt10 really be considered equivalent to Rab5? The authors should clarify how this would work in vivo.

We show that Ypt10 like all other Rab5 homologs in yeast stimulates Mon1-Ccz1 GEF function (Figure 2) and co-localizes with FM4-64 (as an endocytic tracer) and Mon1-Ccz1 itself (Figure 1). We showed before that Vps21 and Ypt52 are most critical for Mon1-Ccz1 and CORVET localization to endosomes (Nordmann et al., 2010; Cabrera et al., 2013). The copy number of Ypt10 (1790 per cell, see *Saccharomyces* genome database) is about 1/10^th^ of the much more abundant Vps21 (8700) and Ypt52 (9750) together, and Ypt53 (1170) is a stress induced isoform (Schmidt et al., 2017; Nickerson et al., 2012). Likewise, Ypt10 could be a stress-induced Rab5 isoform in yeast, and its function thus remains to be clarified. Importantly, we use Ypt10 here as an example to show the effect of Rab5-like proteins in our in vitro assay, yet find similar stimulation of Mon1-Ccz1 with Vps21 (Figure 3) and the *Drosophila* homolog Rab5 (Figure 4).

Therefore, we also consider our findings also to be relevant regarding mammalian cells. If and how it is true for every single member of the entire Rab5-family of mammalian cells or even different tissue has to be addressed in further studies.

One of the more intriguing concepts was the possible regulation by a third subunit, creating a trimeric GEF complex in metazoans and Drosophila. Given the suggestion by Pontano-Vaites et al. that mammalian RMC1 regulates Mon1-Ccz1, it was (mildly) disappointing that the trimeric Drosophila Bulli-Mon1-Ccz1 complex displayed GEF activity similar to that of the dimeric Mon1-Ccz1 complex. The authors cite "Dehnen et al., submitted" for a description of the *Drosophila* system. However, this topic is sufficiently interesting that having access to data in the Dehnen et al. manuscript is crucial. For example, are the sequences of RMC1 and Bulli similar or has the Mon1-Ccz1 complex evolved two alternative pathways for regulation? I do not mean to ask that data of Dehnen et al. manuscript are incorporated in this study, but it should be made available, e.g. in bioRxiv.

We were hoping that this in vivo study would have been published already. It was caught in reviews, revised, and is now under consideration at JCS.

Another aspect of the work which was fascinating is the regulation of Mon1-Ccz1 by the Rab protein Ypt10p. Based on the fact that Ypt10p binds Mon1-Ccz1 with higher affinity than other Rab proteins, one would expect the enhancement of GEF activity to be caused by more Mon1-Ccz1 bound to the liposome. However, the authors showed this was not the case, suggesting an actual regulatory mechanism for Ypt10. Delving into this mechanism more thoroughly would make this story of interest to a wider audience. For example, can soluble Ypt10 stimulate Mon1-Ccz1 activity? It would be nice if the authors could provide some insights into this mechanism. While I don't think it's mandatory for publication, it is at least a topic for further discussion.

The reviewer brings up an important issue – the crosstalk of Rab5 with Mon1-Ccz1. Again, we would like to stress that our point is not so much Ypt10, but Rab5 in general. Ypt10 just has the strongest binding affinity among the yeast Rab5 homologs. We nevertheless consider the question if soluble Ypt10 can stimulate Mon1-Ccz1 activity an important control. We therefore now included a new Figure 2—figure supplement 2, where we addressed this. Soluble Ypt10 was not able to trigger Mon1-Ccz1 GEF-activity, even in a 10-fold excess compared to membrane bound prenylated Ypt10 (and therefore a 150x molar excess compared to Mon1-Ccz1). Delving further into this interaction will need further investigations and is part of future projects.

Reviewer #2:Ungermann and co-workers set out to define the minimal molecular determinants for conversion of Rab5 positive endosomes to Rab7 positive endosomes, a process that occurs in cells during endosome to lysosome maturation. To fully explain this process, we need to understand how Rab5 promotes Rab7 activation, and how Rab7 promotes Rab5 release. The focus of this work is very much on how Rab5 promotes Rab7 activation through the Mon1-Ccz1 GEF complex and how Rab5 interaction with the GEF is modulated by casein kinase. The authors provide a compelling case addressing this first part, and how the Rab5-dependent pool of Mon1-Ccz1 is finally released by casein kinase activity. However, this leaves open the question of how Rab7 feeds back on to Rab5 to promote Rab5 release. Previous literature provides some clues to this, and the authors should adjust their Abstract, manuscript text and model to better reflect this.

The reviewer is right, we leave this question open. However, at this point, our manuscript only addresses the Rab5-dependent activation of Mon1-Ccz1 and we therefore feel that a further extension on this issue would shift the overall balance of the manuscript. As presented, we also include it both in the Introduction and Discussion. At this point, we rather refer to reviews and publications by the Novick, Zerial or Barr labs, including the work on SAND-1 in *C. elegans*, which nicely discuss Rab5 inactivation. We hope the reviewer agrees with this.

I will address the three key questions raised by the authors in their Abstract:i) that Mon1-Ccz1 is an effector of Rab5.This claim is based on observations that Ypt10 can interact with Mon1-Ccz1 and related Rabs. Overall the data is convincing. However, the hypothesis was surely that Ypt51/52/53 recruit Mon1-Ccz1, so there is a need to look at all these proteins together. Figure 1E shows Ypt10 and Ccz1, can the authors show similar data for Vps21 and Ypt52 which have a much greater effect on Ypt7 localisation (Figure 1F)?

The reviewer brings up an important issue – the crosstalk of different Rab5-homologs with Mon1-Ccz1 (see also our response to reviewer #1). We would like to stress that our point is not so much the organization of the endosomal pathway, but an understanding of the transition from Rab5 to Rab7. We included the colocalization of Mon1-Ccz1 with Ypt10 (Figure 1), since the interaction of Ypt10 with Mon1-Ccz1 was a surprising finding in the GST-pulldown experiments, and we had to verify that both proteins indeed colocalize in vivo. We agree that a detailed systematic characterization of the function and the crosstalk of the four yeast Rab5 homologs with Mon1-Ccz1, and their influence on Rab7 activation in vivo is important and will be topic of further investigations in our lab. At this point we therefore only included the analysis of the different multiple deletions (Figure 1—figure supplement 1) and their vacuolar morphology.

Figure 1D and Figure 1—figure supplement 1. Can the authors show the other subunits of the GEF, not only Ccz1. This is necessary to conclude that the GEF complex has bound, not only one subunit. How reproducible is this? Graphed data from multiple experiments would address this.

We appreciate the reviewer’s concern. However, Mon1-Ccz1 interact via a hydrophobic interface that was described in detail in Kiontke et al., 2017. This complex has never dissociated in our hands, and is purified as a 1:1 complex from yeast (Figure 3C). Therefore, decorating for one subunit surely informs on the other being present. As for the Rab interactions – the only tight interaction is seen with Ypt10, where we showed here already two reproducible pull-downs (Figure 1 and Figure 1—figure supplement 2), and have of course repeated the experiment as described. The other interactions of Mon1-Ccz1 with Rab5s by GST-pull down are rather disappointing. However, their functional role becomes clear during the remaining manuscript. We therefore decided to rely on our presented data.

ii) that membrane-bound Rab5 is the key factor to directly promote Mon1-Ccz1 dependent Rab7 activation and Rab7- dependent membrane fusion.Experiments on the interaction of Mon1-Ccz1 with membranes are done with Vps21 and PIPs (Figure 2), when based on the data in Figure 1 Ypt10 would be more appropriate. This becomes important for Figure 3, when the effects on Rab7 GEF activity are tested. Ypt10 shows a robust effect while the others Rab5-related proteins show a more modest stimulation. This is not reflected in the membrane pelleting shown in Figure 3E, where the amount of Mon1-Ccz1 recovered is equivalent in all cases.Figure 5 switches back to Ypt10, and shows very clear evidence that it promotes Mon1-Ccz1 GEF activity and membrane fusion. Does this work for the other Rab5 family members?

We have now included the interaction of Yp10 and Mon1-Ccz1 on PIPs (Figure 2—figure supplement 1), and find the same. We have not repeated the fusion assay with all other Rab5 proteins. Given that we observe robust GEF activity with all Rab5s (Figure 3), and with the *Drosophila* system (Figure 4), we feel that our data sufficiently address this point. The reason for using Ypt10 is simply that it shows the strongest effect. We feel that a detailed comparison is an issue of future studies.

iii) that this process is regulated in yeast by the casein kinase Yck3, which phosphorylates Mon1 and blocks Rab5 binding. Our study thus uncovers the minimal feed-forward machinery of the endosomal Rab cascade and novel regulatory mechanism, which can explain organelle maturation in eukaryotic cells.This conclusion is supported by Figure 6 and the data show Yck3 blocks both Vps21 and Ypt10 interaction, presumably by acting at the same site on the complex. Do the authors have any idea where Yck3 phosphorylates the Mon1-Ccz1 complex? It is unlikely but can we completely exclude the possibility that the Rabs are Yck3 targets?

We thank the reviewer for these questions. We have so far only evidence that Mon1-Ccz1 is a substrate of Yck3, and find phosphorylation sites in both proteins, which we currently map.

We cannot exclude that the Rabs are also modified by Yck3. When we incubate Ypt7 or Vps21 with the kinase Yck3, we do not observe any size shift so far, suggesting that they are not modified. We believe that this issue will be resolved as soon as we know exactly where Rab5 proteins bind Mon1 – an issue we would like to follow up in future studies.

Reviewer #3:[…] I think the in vivo experiments as presented do not currently provide sufficient information needed to properly assess the true physiological relevance of the Ypt10 interaction.1) The authors find that Ypt10 is the best physical interactor with Mon1/Ccz1, yet they have not thoroughly tested its importance in vivo. As shown in Figure 1F, Ypt10 does not appear to be required for Ypt7 activation in cells, but this could be due to redundancy with the other Rab5 paralogs. The authors should determine whether double mutants of ypt10 and vps21 (and perhaps other double-mutant combinations) perturb Ypt7 activation in cells.

As also mentioned to Reviewer 1, it was not our intention to clarify the main functions of Ypt10. This paper is, as the reviewer rightly points out, about Rab5-driven Mon1-Ccz1 activation, and Ypt10 just happens to be the highest affinity interactor among the yeast Rab5-like proteins for Mon1-Ccz1. It otherwise behaves like all Rab5-like proteins in yeast and colocalizes with Mon1-Ccz1 to dots proximal to the vacuole (see also response to reviewer #1), yet Ypt10 seems to be of low abundance. We now point this explicitly out in the manuscript.

2) I'm also surprised the authors did not examine localization of Mon1/Ccz1 in various Rab5 mutants. Shouldn't there be a Mon1/Ccz1 mislocalization phenotype in some combination of Rab5 homolog (double, triple, etc). mutant cells, especially involving ypt10?

The reviewer brings up an important issue. Any yeast mutant along the endocytic pathway that interferes with endosomal biogenesis (we call them Class D mutants) interferes with Mon1Ccz1 localization (Nordmann et al., 2010). It is thus not so simple as redundancy interferes with function (see Cabrera et al., 2013). Moreover, Ypt10 is of low abundance, consistent with the idea that the protein seems to be regulated by cellular stress. For the reviewer’s request, we now included the different combinations of Rab5 deletions and monitor the morphology of vacuoles (Figure 1—figure supplement 1). The effect on Mon1-Ccz1 mislocalization was already seen in the *vps21 ypt52* mutant, but we believe that we learn more about the in vivo function once we have a mutant in Mon1-Ccz1 that blocks the Rab5 interaction and therefore decided to reserve this issue for a future study.

3) Furthermore, how do the authors explain the punctate localization of Ypt7 in vps21-ypt52 double mutant cells? The punctate localization suggests Ypt7 is still being activated in these mutants, and is activated at an endosomal structure rather than on the vacuole. In fact, one interpretation of this observation is that Mon1/Ccz1 localization and function is unperturbed in the vps21-ypt52 double mutants, and instead the maturation process is perturbed.

We agree on this interpretation with the reviewer, but this is a general problem with the analysis of mutants that interfere with the endosomal system in yeast (see earlier work of the Emr group, and also Cabrera et al., 2013). For instance, we are not even sure if the dot localization is due to a very small vacuole and a large endosome or a specific mislocalization of Ypt7.

As pointed out to reviewer 1 and 2, our focus was not on the function of Rab5 and the general organization of endosomes in vivo but on the role of Rab5 in Mon1-Ccz1 recruitment and activation. Further characterization of the interconnection of the different yeast Rab5 homologs and therefore organization of the early endosomal pathway is beyond the scope of this manuscript, and certainly an important issue for future studies.

4) The authors switch which Rab homolog they are examining in different experiments. The authors state that they would use Ypt10 in "further assays", yet in Figure 2 they use Vps21 to test liposome-binding of Mon1/Ccz1. Why wasn't Ypt10 also tested in Figure 2?

See below.

5) Similarly, as I was reading the manuscript, in Figure 2 I was expecting a direct comparison of liposome binding of Mon1/Ccz1 for all of the yeast Rab5 paralogs, as well as at least one negative control (i.e. Ypt6 or Ypt31). The experiments shown in Figure 3, in which the specificity of several different Rabs in stimulating Mon1/Ccz1 GEF activity is tested, could potentially obviate the need for such an experiment. However, in the second paragraph of the subsection “Control of Mon1-Ccz1 GEF activity by Rab5 is evolutionarily conserved” the authors claim that the GEF activity correlates with the affinity of the effector interaction between the Rab and Mon1/Ccz1 – I think in order to make this claim they would need to compare the relative strengths of the different Rabs in recruiting Mon1/Ccz1 to membranes through liposome binding experiments similar to those shown in Figure 2.

Combined answer to points 4 and 5:

We now included data of Ypt10 dependent recruitment of Mon1-Ccz1 to liposomes as a new Figure 2—figure supplement 1. We assumed that using Ypt10 as a strong recruiter in these assays would lead to an underestimation on the influence that lipids have on the recruitment, yet observed the same effect with Vps21, which clearly is the weaker interactor of Mon1-Ccz1 (Figure 1). This suggests that it is rather the crosstalk of the Rab5-proteins with Mon1-Ccz1 that determines the stimulation in GEF activity and not just the membrane binding.

As for Figure 3, we clearly show the k_obs_ values and see that Ypt10 is the strongest stimulator of GEF activity, and it also binds to Mon1-Ccz1 in vitro best (Figure 1). We believe that it is exactly this issue, the membrane binding as a claim of function, which we clarify in our study. Our data clearly demonstrate that membrane binding per se can show cooperation (Figure 2), but depending on the set up can also be meaningless (Figure 3E), unless activity is also monitored, at least for Mon1-Ccz1 as a GEF.